# Unifying Unsupervised Graph-Level Anomaly Detection and Out-of-Distribution Detection: A Benchmark

**Yili Wang**[1,*], **Yixin Liu**[2,*], **Xu Shen**[1,*], **Chenyu Li**[1,*], **Kaize Ding**[3], **Rui Miao**[1],
**Ying Wang**[1], **Shirui Pan**[2,†], **Xin Wang**[1,†]
[1]Jilin University, [2]Griffith University, [3]Northwestern University
```
{wangyl21, shenxu23, chenyul23, ruimiao20}@mails.jlu.edu.cn,
{yixin.liu, s.pan}@griffith.edu.au,
kaize.ding@northwestern.edu,
{wangying2010, xinwang}@jlu.edu.cn
```

## Abstract

To build safe and reliable graph machine learning systems, unsupervised graph-level anomaly detection (GLAD) and unsupervised graph-level out-of-distribution (OOD) detection (GLOD) have received significant attention in recent years. Though those two lines of research indeed share the same objective, they have been studied independently in the community due to distinct evaluation setups, creating a gap that hinders the application and evaluation of methods from one to the other. To bridge the gap, in this work, we present a Unified Benchmark for unsupervised Graph-level OOD and anomaLy Detection (UB-GOLD), a comprehensive evaluation framework that unifies GLAD and GLOD under the concept of generalized graph-level OOD detection. Our benchmark encompasses 35 datasets spanning four practical anomaly and OOD detection scenarios, facilitating the comparison of 18 representative GLAD/GLOD methods. We conduct multi-dimensional analyses to explore the effectiveness, OOD sensitivity spectrum, robustness, and efficiency of existing methods, shedding light on their strengths and limitations. Furthermore, we provide an open-source codebase (https://github.com/UB-GOLD/UB-GOLD) of UB-GOLD to foster reproducible research and outline potential directions for future investigations based on our insights.

## 1 Introduction

With the ubiquity of graph data, graph machine learning has been widely adopted in various scientific and industrial fields, ranging from bioinformatics to social networks (Xia et al., 2021; Wu et al., 2020; Zhang et al., 2024b; Wang et al., 2024b; Zhang et al., 2024a; Wang et al., 2022; Juan et al., 2024; Miao et al., 2024; Liu et al., 2024; Ding et al., 2024a). As one of the representative graph learning tasks, graph-level anomaly detection (GLAD) has been widely studied to identify abnormal graphs that show significant dissimilarity from the majority of graphs in a collection (Ma et al., 2022; Zhang et al., 2022). GLAD task is crucial in various real-world applications, such as toxic molecule recognition and pathogenic brain mechanism discovery (Jiang et al., 2021; Liu et al.). Due to the high cost of data labeling, existing GLAD studies generally follow an unsupervised paradigm, eliminating the requirement of labeled anomaly samples for model training (Ma et al., 2022; Qiu et al., 2022; Liu et al., 2023b; Pan et al., 2023).

In the meantime, another line of research – graph-level out-of-distribution detection (GLOD) – has drawn increasing attention in the research community lately (Liu et al., 2023a; Li et al., 2022; Wu et al., 2024). GLOD aims to identify whether each graph sample in a test set is in-distribution (ID), meaning it comes from the same distribution as the training data, or out-of-distribution (OOD), indicating it comes from different distributions. Considering the universality of distribution shift in open-world data, GLOD plays an important role in real-world high-stakes applications such as drug discovery and cyber-attack detection (Shen et al., 2024; Ji et al., 2023; Ju et al., 2024; Ding et al., 2024b; Li et al., 2024). Though a few post-hoc GLOD methods (Guo et al., 2023; Wang et al.,

---
[*]Equal Contribution.
[†]Corresponding Authors.

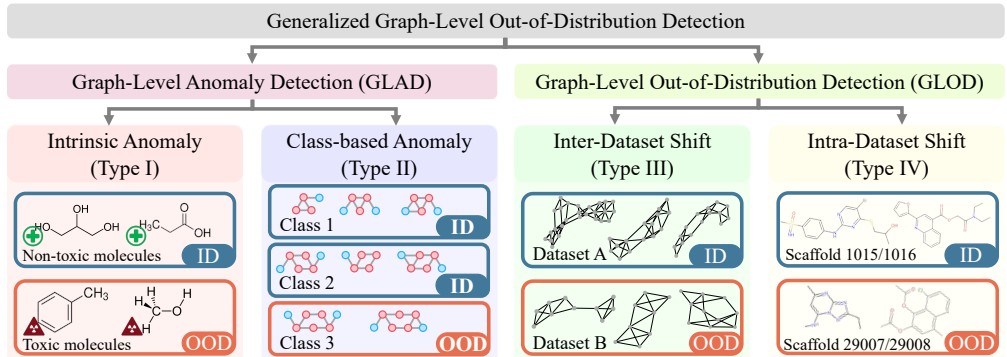

Figure 1: Diagram of generalized OOD detection scenarios supported by UB-GOLD.

2024a) that rely on a well-trained graph classifier have been proposed, most of the existing GLOD methods (Liu et al., 2023a; Li et al., 2022; Shen et al., 2024) are developed in an unsupervised fashion, where a specific OOD detection model is trained on unlabeled ID data, and then predicts a score for each test sample to indicate its ID/OOD status. In essence, unsupervised GLOD and unsupervised GLAD share the same goal, suggesting that research in one area could potentially be applied to solve problems in the other. Recent survey and benchmark papers (Yang et al., 2022; 2021) also identify anomaly detection and OOD detection as two branches under the concept of "generalized OOD detection", which further highlights the close conceptual connection between GLOD and GLAD tasks.

Despite the inherent connection between the problems of unsupervised GLAD and GLOD, those two research sub-areas have been studied independently in the literature and have distinct evaluation setups. To fill the gap, in this paper, we develop a **U**nified **B**enchmark for unsupervised **G**raph-level **O**OD and anoma**L**y **D**etection (UB-GOLD for short). As shown in Fig. 1, we unify GLAD and GLOD as "generalized graph-level OOD detection problem", and consider two GLOD scenarios and two GLAD scenarios for comprehensive evaluation. For GLAD task that aims to detect graph samples with semantic abnormality, we explore scenarios with *intrinsic anomaly* and *class-based anomaly*. Specifically, the first type of dataset inherently includes semantic anomalies, while the second type treats samples from one class (usually the minority class) as anomalies. For GLOD task that emphasizes the distribution shift between ID and OOD samples, we consider two scenarios with different distribution shifts, i.e., *inter-dataset shift* and *intra-dataset shift*. Specifically, the former simulates distribution shifts by drawing ID and OOD samples from different datasets within the same domain (Liu et al., 2023a; Sehwag et al., 2021), while the latter refers to datasets with intrinsic distribution shifts regarding graph sizes or molecular scaffolds (Gui et al., 2022). Based on 35 datasets belonging to 4 types, we establish a comprehensive benchmark to fairly compare 18 GLAD and GLOD methods under a unified experimental setting, exploring the **effectiveness** of state-of-the-art (SOTA) approaches across diverse scenarios and domains.

Apart from comparing performance, we further investigate the characteristics of GLAD/GLOD methods in terms of three dimensions: OOD sensitivity spectrum, robustness, and efficiency, through extensive experiments. For **OOD sensitivity spectrum**, we test a GLAD/GLOD model on OOD data with varying degrees of distribution shift, investigating each method's ability to identify both near-OOD and far-OOD samples. For **robustness**, we introduce varying proportions of OOD samples into the training data to observe the impact of noisy ID data on the performance of existing methods. For **efficiency**, we perform efficiency evaluations for representative GLAD/GLOD approaches, focusing on time and space complexity. At the end of this paper, we discuss the future directions of this emerging research direction. To sum up, the main contribution of this paper is three-fold:

- **Comprehensive benchmark.** We introduce UB-GOLD, the first comprehensive and unified benchmark for GLAD and GLOD. UB-GOLD compares 18 representative GLAD/GLOD methods across 35 datasets from four practical anomaly and OOD detection scenarios.
- **Multi-dimensional analysis.** To explore their ability and limitations, we conduct a systematic analysis of existing methods from multiple dimensions, encompassing effectiveness on different datasets, OOD sensitivity spectrum to near-OOD and far-OOD, robustness against unreliable training data, and efficiency in terms of time and memory usage.
- **Unified Codebase and future directions.** To facilitate future research and quick implementation, we provide an open-source codebase for UB-GOLD. We also outline potential directions based on our findings to inspire future research.

**Key findings.** Through comprehensive comparison and analyses, we have the following remarkable observations: 1) The SOTA GLAD/GLOD methods show excellent performance across tasks; 2) Near-OOD samples are harder to detect compared to far-OOD samples; 3) Most methods are vulnerable to the perturbations of training sets; and 4) Certain end-to-end methods outperform two-step methods in terms of both performance and computational costs.

## 2 RELATED WORK AND PROBLEM DEFINITION

In this section, we begin by briefly reviewing related work on GLAD and GLOD. Next, we define the generalized unsupervised graph-level OOD detection problem, formulating GLAD and GLOD into a unified learning task.

**Graph-level anomaly detection (GLAD).** GLAD aims to identify abnormal graphs that show significant dissimilarity from the majority of graphs in a collection. A simple solution for GLAD is to use graph kernels (Vishwanathan et al., 2010; Neumann et al., 2016) to extract graph-level features and utilize shallow anomaly detectors (e.g. OCSVM (Amer et al., 2013) and iForest (Liu et al., 2008)) to identify anomalies. Recently, GNN-based end-to-end methods have demonstrated significant performance improvements in GLAD (Ma et al., 2022; Zhao & Akoglu, 2023). Among them, some approaches assume that the anomaly labels of graphs are available for model training, and hence formulate GLAD as a supervised classification problem (Zhang et al., 2022; Ma et al., 2023; Dong et al., 2024). Nevertheless, their heavy reliance on labeled anomalies makes it challenging to apply them to real-world scenarios where annotated anomalies are scarce or unavailable. To overcome this shortage, the majority of GNN-based GLAD methods focus on an unsupervised learning paradigm, learning the GLAD model only from normal data (Ma et al., 2022; Qiu et al., 2022; Liu et al., 2023b; Zhao & Akoglu, 2023; Luo et al., 2022; Li et al., 2023). Recent research (Liu et al., 2023a; Li et al., 2022) indicates that GLAD methods also show great potential in handling GLOD problem, prompting us to unify these two fields into a generalized research problem and conduct a comprehensive benchmarking study.

**Graph-level out-of-distribution detection (GLOD).** GLOD refers to the problem of identifying whether a graph sample in a test set is ID or OOD, i.e., whether it originates from the same distribution as the training data or from a different one (Li et al., 2022). A line of studies developed post-hoc OOD detectors that identify OOD samples by additional fine-tuned detectors on top of well-trained GNN classifiers (Guo et al., 2023; Wang et al., 2024a). However, these methods require annotated ID data to train the backbone GNNs, which limits their applicability in scenarios where labeled data is unavailable. In contrast, another line of research proposes training an OOD-specific GNN model using only ID data, without relying on any labels or OOD data (Shen et al., 2024; Liu et al., 2023a; Li et al., 2022). To learn discriminative patterns of ID data, they employ unsupervised learning techniques such as contrastive learning (Liu et al., 2023a) and diffusion model (Shen et al., 2024).

**Problem definition.** Inspired by generalized OOD detection framework in computer vision (Yang et al., 2021; Gui et al., 2022), and due to the highly similar objectives of GLAD and GLOD (for detailed discussion, see Appendix D.1), we unify GLAD and GLOD into a high-level topic termed **generalized graph-level OOD detection**. The new learning task aims to distinguish generalized OOD samples (i.e., OOD or abnormal graphs) from generalized ID samples (i.e., ID or normal graphs). In this paper, we focus on the unsupervised scenario, considering its universality. The research problem is formulated as:

**Definition 1.** [Unsupervised generalized graph-level OOD detection] We define the training dataset as $\mathcal{D}_{train} = \{G_1, \cdots, G_n\}$, where each sample $G_i$ is a graph drawn from a specific distribution $\mathbb{P}^{in}$. We define the testing dataset as $\mathcal{D}_{test} = \mathcal{D}_{test}^{in} \cup \mathcal{D}_{test}^{out}$, where $\mathcal{D}_{test}^{in}$ contains graphs sampled from $\mathbb{P}^{in}$, and $\mathcal{D}_{test}^{out}$ consists of graphs drawn from an OOD distribution $\mathbb{P}^{out}$. Given $\mathcal{D}_{train}$ and $\mathcal{D}_{test}$, the learning objective is to train a detection model $f(\cdot)$ on $\mathcal{D}_{train}$ and then the model can predict whether each sample $G' \in \mathcal{D}_{test}$ belongs to $\mathbb{P}^{in}$ or $\mathbb{P}^{out}$. In practice, $f(\cdot)$ is a scoring function, where a larger OOD score $s' = f(G')$ indicates a higher probability that $G'$ is from $\mathbb{P}^{out}$ (i.e., abnormal samples).

$$g(G'; \tau, f) = \begin{cases} 0 \text{ (OOD)}, & \text{if } f(G') \leq \tau, \\ 1 \text{ (ID)}, & \text{if } f(G') > \tau. \end{cases} \tag{1}$$

The theoretical objective of the scoring function $f(G')$ is to maximize the separation between the distributions of ID/normal graphs and OOD/anomalous graphs:

$$\max_f \mathbb{E}_{G' \sim \mathcal{D}_{test}^{in}} f(G') - \mathbb{E}_{G' \sim \mathcal{D}_{test}^{out}} f(G'). \tag{2}$$

Table 1: Statistics of datasets used in UB-GOLD.

| Dataset Type | Full Name | Abbreviation | Domain | OOD Definition | # ID Train | # ID Test | # OOD Test | # Anomaly Ratio % |
|---|---|---|---|---|---|---|---|---|
| **(Type I)** Intrinsic Anomaly | Tox21_p53 | p53 | Molecules | Inherent Anomaly | 8088 | 241 | 28 | 0.34 |
| | Tox21_HSE | HSE | Molecules | Inherent Anomaly | 423 | 257 | 10 | 1.45 |
| | Tox21_MMP | MMP | Molecules | Inherent Anomaly | 6170 | 200 | 38 | 0.59 |
| | Tox21_PPAR-gamma | PPAR | Molecules | Inherent Anomaly | 219 | 252 | 15 | 3.09 |
| **(Type II)** Class-based Anomaly | COLLAB | - | Social Networks | Unseen Classes | 1920 | 480 | 520 | 17.81 |
| | IMDB-BINARY | IMDB-B | Social Networks | Unseen Classes | 400 | 100 | 100 | 16.67 |
| | REDDIT-BINARY | REDDIT-B | Social Networks | Unseen Classes | 800 | 200 | 200 | 16.67 |
| | ENZYMES | - | Proteins | Unseen Classes | 400 | 100 | 20 | 3.85 |
| | PROTEINS | - | Proteins | Unseen Classes | 360 | 90 | 133 | 22.81 |
| | DD | - | Proteins | Unseen Classes | 390 | 97 | 139 | 22.20 |
| | BZR | - | Molecules | Unseen Classes | 69 | 17 | 64 | 42.67 |
| | AIDS | - | Molecules | Unseen Classes | 1280 | 320 | 80 | 4.76 |
| | COX2 | - | Molecules | Unseen Classes | 81 | 21 | 73 | 41.71 |
| | NCI1 | - | Molecules | Unseen Classes | 1646 | 411 | 411 | 16.65 |
| | DHFR | - | Molecules | Unseen Classes | 368 | 93 | 59 | 11.35 |
| **(Type III)** Inter-Dataset Shift | IMDB-MULTI→IMDB-BINARY | IM→IB | Social Networks | Unseen Datasets | 1350 | 150 | 150 | 9.09 |
| | ENZYMES→PROTEINS | EN→PR | Proteins | Unseen Datasets | 540 | 60 | 60 | 9.09 |
| | AIDS→DHFR | AI→DH | Molecules | Unseen Datasets | 1800 | 200 | 200 | 9.09 |
| | BZR→COX2 | BZ→CO | Molecules | Unseen Datasets | 364 | 41 | 41 | 9.19 |
| | ESOL→MUV | ES→MU | Molecules | Unseen Datasets | 1015 | 113 | 113 | 9.11 |
| | TOX21→SIDER | TO→SI | Molecules | Unseen Datasets | 7047 | 784 | 784 | 9.10 |
| | BBBP→BACE | BB→BA | Molecules | Unseen Datasets | 1835 | 204 | 204 | 9.09 |
| | PTC_MR→MUTAG | PT→MU | Molecules | Unseen Datasets | 309 | 35 | 35 | 9.23 |
| | FREESOLV→TOXCAST | FS→TC | Molecules | Unseen Datasets | 577 | 65 | 65 | 9.19 |
| | CLINTOX→LIPO | CL→LI | Molecules | Unseen Datasets | 1329 | 148 | 148 | 9.11 |
| **(Type IV)** Intra-Dataset Shift | GOOD-HIV-Size | HIV-Size | Molecules | Size | 1000 | 500 | 500 | 25.00 |
| | GOOD-ZINC-Size | ZINC-Size | Molecules | Size | 1000 | 500 | 500 | 25.00 |
| | GOOD-HIV-Scaffold | HIV-Scaffold | Molecules | Scaffold | 1000 | 500 | 500 | 25.00 |
| | GOOD-ZINC-Scaffold | ZINC-Scaffold | Molecules | Scaffold | 1000 | 500 | 500 | 25.00 |
| | DrugOOD-IC50-Size | IC50-Size | Molecules | Size | 1000 | 500 | 500 | 25.00 |
| | DrugOOD-EC50-Size | EC50-Size | Molecules | Size | 1000 | 500 | 500 | 25.00 |
| | DrugOOD-IC50-Scaffold | IC50-Scaffold | Molecules | Scaffold | 1000 | 500 | 500 | 25.00 |
| | DrugOOD-EC50-Scaffold | EC50-Scaffold | Molecules | Scaffold | 1000 | 500 | 500 | 25.00 |
| | DrugOOD-IC50-Assay | IC50-Assay | Molecules | Protein Target | 1000 | 500 | 500 | 25.00 |
| | DrugOOD-EC50-Assay | EC50-Assay | Molecules | Protein Target | 1000 | 500 | 500 | 25.00 |

This unified objective integrates the detection principles of GLAD and GLOD, emphasizing their shared goal of distinguishing graphs that deviate from a given distribution.

# 3 BENCHMARK DESIGN

In this section, we provide a comprehensive overview of UB-GOLD, covering datasets (Sec. 3.1), algorithms (Sec. 3.2), and evaluation metrics & implementation details (Sec. 3.3).

## 3.1 BENCHMARK DATASETS

UB-GOLD includes 35 datasets in four types, each of which corresponds to either a GLAD or GLOD scenario categorized in Fig.1. Table 1 provides an overview of the benchmark datasets. For **GLAD task**, we consider two types of datasets with intrinsic anomaly and class-based anomaly to test the models' performance of detecting anomalous graphs. For **GLOD task**, we consider datasets with inter-dataset shift and intra-dataset shift that assess the models' ability to distinguish between ID and OOD samples. For more detailed information on the datasets, please see Appendix B.

- **Type I: Datasets with intrinsic anomaly.** These datasets contain natural anomalies within chemical compounds, testing the robustness of GLAD methods. We use four datasets (i.e., Tox21_HSE, Tox21_MMP, Tox21_p53, and Tox21_PPAR-gamma) from the Tox21 challenge (Abdelaziz et al., 2016) involving molecules with unexpected biological activities.

- **Type II: Datasets with class-based anomaly.** In these datasets, certain classes are designated as anomalies. We use 11 datasets (e.g., COLLAB, IMDB-BINARY, and ENZYMES) from the TU benchmark (Morris et al., 2020) where minority or distinct class samples are treated as anomalies.

- **Type III: Datasets with inter-dataset shift.** We synthetic a benchmark dataset with inter-dataset shift by considering samples from one real-world dataset as ID and samples from another real-world dataset as OOD. For example, in "IMDB-MULTI→IMDB-BINARY", graphs from IMDB-MULTI are ID, and graphs from IMDB-BINARY are OOD. The datasets belonging to the same domain and having close distribution shifts form a pair (Liu et al., 2023a).

- **Type IV: Datasets with intra-dataset shift.** Designed to assess GLOD methods under various types of intra-dataset shifts, these datasets are from graph OOD benchmarks, including GOOD (Gui et al., 2022) and DrugOOD (Ji et al., 2023). Specifically, datasets with three kinds of domain shifts, i.e., assay shift, scaffold shift, and size shift, are considered.

Table 2: Categorization of all benchmark algorithms in UB-GOLD.

| Method Type | Category | Models |
|---|---|---|
| Two-Step | Graph kernel with detector (GK-D) | PK-SVM, PK-iF, WL-SVM, WL-iF |
| | Self-supervised learning with detector (SSL-D) | IG-SVM, IG-iF, GCL-SVM, GCL-iF |
| End-to-End | Graph neural network-based GLAD | OCGIN, GLADC, GLocalKD, OCGTL, SIGNET, CVTGAD |
| | Graph neural network-based GLOD | GOOD-D, GraphDE, AAGOD, GOODAT |

## 3.2 BENCHMARK ALGORITHMS

In UB-GOLD, we consider four groups of GLAD/GLOD methods for a comprehensive evaluation: 1) two-step methods, including graph kernel with detector (GK-D) and self-supervised learning (SSL) with detector (SSL-D), and 2) End-to-End methods, including GNN-based GLAD and GLOD methods. For the end-to-end methods, we further divided them into 4 categories from the technical perspective, including one-class classification-based, graph reconstruction-based, contrastive learning-based, and well-trained GNN-based. All methods are unsupervised, aligning with the primary focus of UB-GOLD to provide a comprehensive and fair evaluation framework. For a detailed summary of the models and further details, please refer to Table 2 and Appendix C.

- **Graph kernel with detector.** These methods follow a two-step process: obtaining graph embeddings using graph kernels and applying outlier detectors in the embedding space. For kernel methods, we consider Weisfeiler-Leman subtree kernel (WL) (Li et al., 2016) and propagation kernel (PK) (Neumann et al., 2016). For detectors, we employ isolation forest (iF) (Liu et al., 2008) and one-class SVM (OCSVM, SVM for short) (Amer et al., 2013).

- **SSL with detector.** These approaches also follow a two-step process but use SSL methods to obtain graph embeddings. We consider two SSL methods, GraphCL (GCL for short) (You et al., 2020) and InfoGraph (IG for short) (Sun et al., 2020) to generate embeddings and use iF (Liu et al., 2008) and SVM (Amer et al., 2013) as detectors.

- **GNN-based GLAD methods.** We consider 6 SOTA methods for GLAD, including OCGIN (Zhao & Akoglu, 2023), GLocalKD (Ma et al., 2022), OCGTL (Qiu et al., 2022), SIGNET (Liu et al., 2023b), GLADC (Luo et al., 2022), and CVTGAD (Li et al., 2023). These methods use different techniques, such as deep one-class classification, contrastive learning, and knowledge distillation, to detect anomalies in graph data.

- **GNN-based GLOD methods.** We involve four representatives of unsupervised GLOD methods, i.e., GOOD-D (Liu et al., 2023a), GraphDE (Li et al., 2022), AAGOD Guo et al., 2023, and GOODAT (Wang et al., 2024a) for comparison. GOOD-D is a contrastive learning-based approach, while GraphDE is a generative model-based approach. Both AAGOD and GOODAT operate on well-trained GNNs[1]; AAGOD follows a data-centric approach, and GOODAT focuses on test-time OOD detection.

**UB-GOLD: A Unified Benchmark Library for Unsupervised GLAD and GLOD**. We offer our developed benchmark library UB-GOLD, as illustrated in Fig. 2. Specifically, the benchmark starts with a **Benchmark Datasets** module, which includes four types of datasets: intrinsic anomaly, class-based anomaly, inter-dataset shift, and intra-dataset shift. Each dataset type is designed to cover both GLAD and GLOD scenarios, ensuring broad applicability across different graph-level anomaly and OOD detection tasks. The **Benchmark Tasks** module prepares the settings (e.g., datasets splits and pre-processing) for different benchmarking tasks of anomaly and OOD detection. It supports three main tasks: general anomaly/OOD detection, near-far OOD detection, and perturbation-based detection, enabling models to handle different levels of OOD difficulty and assess robustness to data perturbations. This step ensures the data is well-prepared for downstream tasks. Next, in the **Method** module, UB-GOLD includes both two-step methods and end-to-end methods. Specifically, we categorize end-to-end GLAD and GLOD methods into four technical groups: one-class classification, graph reconstruction, well-trained GNNs, and contrastive learning. Finally, the **Evaluation** module ensures a comprehensive and fair assessment, offering Comprehensive Metrics, Robustness under

---

[1]In this context, "well-trained GNNs" refers to graph neural networks that have been pre-trained using a self-supervised or task-specific pretraining strategy, incorporating domain knowledge to effectively learn graph representations. These models are then fine-tuned for the specific GLAD or GLOD tasks.

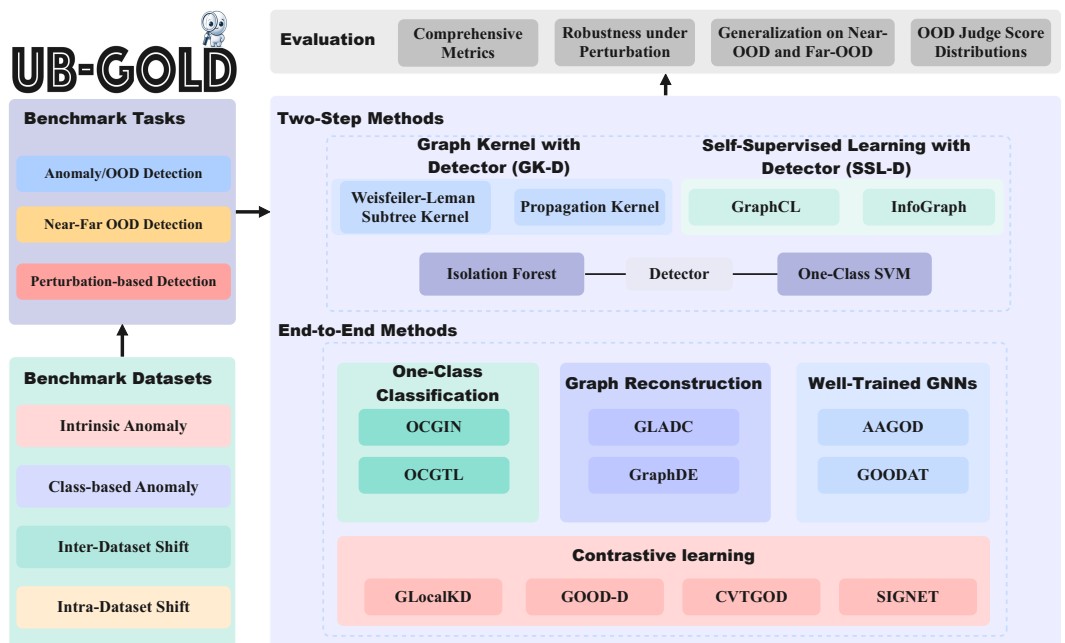

Figure 2: An overview of UB-GOLD.

Perturbation, Generalization on Near-OOD and Far-OOD, and OOD Judge Score Distribution Analysis to measure how well models distinguish between ID and OOD samples.

## 3.3 EVALUATION METRICS AND IMPLEMENTATION DETAILS

**Evaluation metrics.** For comprehensive comparison, UB-GOLD utilizes three commonly used metrics for anomaly/OOD detection (Zhang et al., 2023), i.e., **AUROC**, **AUPRC**, and **FPR95**. Higher AUROC and AUPRC values indicate better performance, while lower FPR95 values are preferable.

**Data split.** In our target scenarios (i.e., unsupervised GLAD/GLOD), all the samples in the training set are normal/ID, while the anomaly/OOD samples only occur in the testing set. In such an unsupervised case, the validation set with anomaly/OOD samples is usually unavailable during the training phase. Thus, following the implementation of OpenOOD (Zhang et al., 2023), we divide the datasets into training and testing sets, without using a validation set. Specifically, we adopted the splits from (Liu et al., 2023a) and (Li et al., 2022), applying them to the benchmark datasets. Detailed splits are provided in Table 1.

**Hyperparameter search.** To obtain the performance upper bounds of various methods on GLAD/GLOD tasks, we conduct a random search to find the optimal hyperparameters w.r.t. their performance on the testing set. The search space is detailed in Table 4. The random search is conducted 20 times or for a maximum of one day per method per dataset to ensure fairness.

For more details related to the experimental setup in UB-GOLD, please refer to Appendix D.

## 4 EXPERIMENTAL RESULTS & DISCUSSION

In this section, we introduce the experimental setup and discuss the experimental results in UB-GOLD benchmark. Specifically, we aim to answer the following research questions: ● **RQ1 (Effectiveness)**: How do different GLAD and GLOD methods perform under various anomaly scenarios and distribution shifts? (Sec. 4.1) ● **RQ2 (OOD sensitivity spectrum)**: How effective are GLAD and GLOD methods in detecting near-OOD and far-OOD graph samples? (Sec. 4.2) ● **RQ3 (Robustness)**: How does the inclusion of OOD samples in the ID training set affect the robustness of GLAD and GLOD methods? (Sec. 4.3) ● **RQ4 (Efficiency)**: Are the GLAD and GLOD methods efficient in terms of run time and memory usage? (Sec. 4.4)

Table 3: Comparison in terms of AUROC. The best three results are highlighted by 1st , 2nd , and 3rd . "Avg. AUROC" and "Avg. Rank" indicate the average AUROC and rank across all datasets.

| | GK-D (two-step) | | | | SSL-D (two-step) | | | | GNN-based GLAD (end-to-end) | | | | | | GNN-based GLOD (end-to-end) | | | |
|---|---|---|---|---|---|---|---|---|---|---|---|---|---|---|---|---|---|---|
| | PK-SVM | PK-iF | WL-SVM | WL-iF | IG-SVM | IG-iF | GCL-SVM | GCL-iF | OCGIN | GLocalKD | OCGTL | SIGNET | GLADC | CVTGAD | GOOD-D | GraphDE | AAGOD | GOODAT |
| p53 | 49.17 | 54.05 | 57.69 | 54.40 | 68.11 | 60.85 | 68.61 | 64.60 | 68.35 | 65.43 | 67.58 | 68.10 | 65.82 | 69.40 | 67.82 | 62.59 | 50.12 | 61.71 |
| HSE | 60.72 | 56.49 | 63.27 | 52.98 | 60.33 | 22.77 | 67.40 | 63.95 | 71.42 | 60.21 | 63.36 | 64.56 | 61.37 | 70.52 | 68.71 | 62.47 | 54.60 | 61.99 |
| MMP | 51.03 | 49.95 | 55.50 | 51.98 | 57.72 | 52.58 | 69.91 | 71.31 | 69.37 | 68.12 | 67.51 | 71.23 | 70.03 | 70.58 | 71.41 | 60.12 | 62.92 | 67.49 |
| PPAR | 53.74 | 48.42 | 57.76 | 49.45 | 61.78 | 63.22 | 68.37 | 69.88 | 67.75 | 65.29 | 66.43 | 68.88 | 69.43 | 68.83 | 68.21 | 66.31 | 57.90 | 66.95 |
| COLLAB | 49.72 | 51.38 | 54.62 | 51.41 | 36.47 | 38.18 | 44.91 | 45.44 | 60.58 | 51.85 | 48.13 | 72.45 | 54.32 | 71.01 | 69.34 | 46.77 | 50.14 | 44.91 |
| IMDB-B | 51.75 | 52.83 | 52.98 | 51.79 | 40.89 | 45.64 | 68.00 | 63.88 | 61.47 | 53.31 | 65.27 | 70.12 | 65.94 | 69.82 | 66.68 | 59.25 | 58.43 | 65.46 |
| REDDIT-B | 48.36 | 46.19 | 49.50 | 49.84 | 60.32 | 52.51 | 84.49 | 82.64 | 82.10 | 80.32 | 89.92 | 85.24 | 78.87 | 87.43 | 89.43 | 63.42 | 68.78 | 80.31 |
| ENZYMES | 52.45 | 49.82 | 53.75 | 51.03 | 60.97 | 53.94 | 62.73 | 63.09 | 62.44 | 61.75 | 63.59 | 63.12 | 63.44 | 68.56 | 64.58 | 52.10 | 58.70 | 52.33 |
| PROTEINS | 49.43 | 61.24 | 53.85 | 65.75 | 61.15 | 52.78 | 72.61 | 72.60 | 76.46 | 77.29 | 72.89 | 75.86 | 77.43 | 76.49 | 76.02 | 68.81 | 75.04 | 75.92 |
| DD | 47.69 | 75.29 | 47.98 | 70.49 | 70.33 | 42.67 | 76.43 | 65.41 | 79.08 | 80.76 | 77.76 | 74.53 | 76.54 | 78.84 | 79.91 | 60.49 | 74.00 | 77.62 |
| BZR | 46.67 | 59.08 | 51.16 | 51.71 | 41.50 | 45.42 | 68.93 | 67.81 | 69.13 | 68.55 | 51.89 | 80.79 | 68.23 | 77.69 | 73.28 | 65.94 | 64.52 | 64.77 |
| AIDS | 50.93 | 52.01 | 52.56 | 61.42 | 87.20 | 97.96 | 95.44 | 98.80 | 96.89 | 96.93 | 97.60 | 98.02 | 99.21 | | 97.10 | 70.82 | 86.64 | 98.82 |
| COX2 | 52.15 | 52.48 | 53.34 | 49.56 | 49.11 | 48.61 | 59.68 | 59.38 | 57.81 | 58.93 | 59.81 | 72.35 | 64.13 | 64.36 | 63.19 | 54.73 | 51.86 | 59.99 |
| NCI1 | 51.39 | 50.22 | 54.18 | 50.41 | 45.11 | 61.88 | 43.33 | 46.44 | 69.46 | 65.29 | 75.75 | 74.32 | 68.32 | 69.13 | 61.58 | 58.74 | 49.94 | 45.96 |
| DHFR | 48.31 | 52.79 | 50.30 | 51.64 | 45.58 | 63.15 | 58.21 | 57.01 | 61.09 | 61.79 | 59.82 | 72.87 | 61.25 | 63.23 | 64.48 | 53.23 | 63.93 | 61.52 |
| IM→IB | 49.80 | 51.23 | 53.45 | 53.03 | 56.26 | 51.32 | 74.45 | 78.62 | 80.98 | 81.25 | 66.73 | 71.10 | 78.28 | 80.23 | 80.94 | 52.67 | 82.17 | 77.66 |
| EN→PR | 52.53 | 53.36 | 53.92 | 51.90 | 46.01 | 33.52 | 59.76 | 63.23 | 61.77 | 59.36 | 67.18 | 62.42 | 56.95 | 64.31 | 63.84 | 54.48 | 50.17 | 64.61 |
| AI→DH | 51.18 | 51.69 | 52.28 | 50.95 | 84.33 | 83.27 | 97.11 | 98.48 | 95.68 | 94.33 | 98.95 | 96.82 | 95.42 | 99.10 | 99.27 | 94.58 | 94.90 | 93.95 |
| BZ→CO | 43.34 | 52.43 | 49.76 | 52.16 | 64.29 | 64.65 | 78.98 | 76.01 | 87.27 | 80.55 | 81.86 | 89.11 | 83.21 | 96.32 | 95.16 | 65.26 | 77.44 | 80.97 |
| ES→MU | 53.99 | 52.63 | 52.13 | 52.28 | 58.12 | 51.57 | 78.66 | 79.66 | 86.70 | 90.55 | 88.32 | 91.43 | 89.30 | 92.41 | 91.98 | 75.65 | 91.91 | 83.92 |
| TO→SI | 53.73 | 51.87 | 53.50 | 52.25 | 64.32 | 66.53 | 66.85 | 64.85 | 67.29 | 69.80 | 68.91 | 66.72 | 72.51 | 68.24 | 66.70 | 72.34 | 66.90 | 67.21 |
| BB→BA | 54.15 | 53.11 | 54.62 | 53.48 | 63.27 | 32.37 | 69.18 | 67.33 | 78.83 | 77.69 | 78.93 | 89.88 | 79.07 | 80.17 | 81.44 | 55.69 | 72.00 | 75.94 |
| PT→MU | 51.52 | 55.87 | 54.03 | 52.12 | 55.88 | 53.78 | 78.10 | 79.74 | 79.27 | 77.54 | 62.51 | 84.63 | 80.12 | 79.44 | 82.05 | 58.28 | 67.65 | 80.42 |
| FS→TC | 50.06 | 54.76 | 51.98 | 53.24 | 44.98 | 49.57 | 67.05 | 66.01 | 66.98 | 68.92 | 64.38 | 78.12 | 67.32 | 69.89 | 71.58 | 60.12 | 67.65 | 67.09 |
| CL→LI | 50.85 | 51.74 | 52.66 | 51.54 | 55.62 | 56.45 | 59.65 | 54.17 | 61.21 | 58.31 | 59.30 | 72.15 | 63.42 | 70.21 | 69.28 | 50.79 | 53.08 | 60.93 |
| HIV-Size | 48.94 | 49.96 | 66.11 | 45.10 | 31.39 | 32.67 | 26.73 | 35.80 | 38.55 | 42.94 | 96.34 | 91.86 | 47.56 | 56.23 | 74.12 | 68.31 | 41.83 | 29.21 |
| ZINC-Size | 48.66 | 50.12 | 50.58 | 48.96 | 53.07 | 53.47 | 52.61 | 53.46 | 54.22 | 54.21 | 57.29 | 52.53 | 58.28 | 52.51 | 68.43 | 55.63 | 56.56 | 52.51 |
| HIV-Scaffold | 49.36 | 48.43 | 44.72 | 54.57 | 58.78 | 58.74 | 61.00 | 59.26 | 56.82 | 54.38 | 58.05 | 70.93 | 63.23 | 59.49 | 62.28 | 53.48 | 60.75 | 55.75 |
| ZINC-Scaffold | 51.12 | 46.66 | 51.17 | 53.28 | 54.77 | 55.17 | 54.04 | 55.80 | 50.12 | 56.79 | 59.24 | 55.79 | 55.73 | | 56.39 | 55.62 | 52.06 | 48.68 |
| IC50-Size | 67.08 | 59.35 | 90.76 | 52.49 | 32.61 | 36.15 | 24.44 | 30.96 | 42.58 | 41.29 | 97.36 | 81.43 | 39.63 | 50.37 | 50.71 | 45.24 | 40.57 | 27.78 |
| EC50-Size | 70.50 | 59.33 | 92.29 | 49.36 | 29.71 | 32.60 | 22.83 | 27.68 | 41.37 | 39.42 | 97.74 | 77.12 | 39.23 | 55.64 | 56.58 | 59.49 | 45.92 | 32.97 |
| IC50-Scaffold | 66.33 | 60.95 | 88.49 | 58.56 | 36.96 | 38.67 | 34.60 | 38.82 | 44.56 | 42.97 | 96.00 | 77.58 | 38.43 | 57.96 | 56.62 | 59.68 | 49.79 | 36.34 |
| EC50-Scaffold | 64.95 | 62.78 | 86.03 | 55.27 | 27.88 | 30.36 | 31.28 | 32.35 | 51.92 | 48.43 | 94.18 | 74.65 | 40.98 | 66.52 | 58.29 | 54.24 | 45.85 | 41.86 |
| IC50-Assay | 54.47 | 53.13 | 59.18 | 52.11 | 51.23 | 51.42 | 51.15 | 51.93 | 52.61 | 49.87 | 68.78 | 65.99 | 52.12 | 54.25 | 53.74 | 57.11 | 48.74 | 45.34 |
| EC50-Assay | 49.08 | 46.66 | 48.43 | 45.32 | 47.80 | 47.81 | 47.80 | 46.02 | 56.11 | 52.44 | 69.31 | 82.97 | 54.34 | 66.51 | 65.57 | 58.96 | 60.42 | 55.79 |
| Avg. AUROC | 52.69 | 53.67 | 57.56 | 52.91 | 52.11 | 50.35 | 61.29 | 61.50 | 66.00 | 64.29 | 73.14 | 75.86 | 65.50 | 71.07 | 71.05 | 60.38 | 61.54 | 61.91 |
| Avg. Rank | 13.80 | 13.54 | 12.03 | 13.86 | 14.00 | 13.83 | 10.83 | 10.63 | 7.26 | 8.71 | 5.43 | 3.37 | 7.03 | 3.69 | 4.14 | 10.58 | 9.89 | 9.43 |

These research questions are designed to comprehensively evaluate the performance, OOD sensitivity spectrum, robustness, and efficiency of GLAD and GLOD methods. By examining these aspects, we aim to provide a thorough understanding of their strengths and limitations across different scenarios. For more detailed experimental descriptions and settings, please refer to Appendix F.

## 4.1 PERFORMANCE COMPARISON (RQ1)

**Experiment design.** We conduct a comprehensive comparison of the detection performance in terms of AUROC, AUPR, and FPR95 of 18 benchmark algorithms on 35 benchmark datasets. For each method and dataset, we conduct 5 runs of experiments and report the average performance.

**Experimental results.** Table 3 shows the performance comparison in terms of AUROC, and Fig. 3 provides box plots to overview the model performance across 35 datasets for all metrics. In Appendix F, detailed performance with standard deviation is demonstrated in Table 8, and the box plots of the ranking of each method are demonstrated in Fig. 7. We have the following observations.

**Observation ❶: The SOTA GLAD/GLOD methods show excellent performance on both tasks**. The results in Table 3 highlight the excellent performance of the SOTA GLAD/GLOD methods on both detection tasks. Specifically, the GLAD methods, SIGNET, CVTGAD, and OCGTL, demonstrate outstanding performance in OOD detection tasks, achieving competitive results in several datasets. Meanwhile, the GLOD method GOOD-D, while not attaining the best performance in anomaly detection tasks, still performs commendably with an average ranking of 4.14, placing it among the top performers. This observation highlights the intercommunity between GLAD and GLOD tasks, emphasizing the need to apply powerful methods designed for one task to the other.

**Observation ❷: No universally superior method.** Table 3 demonstrates that no single method consistently outperforms others on more than 12 (out of 35) datasets. Even the top-ranked method, SIGNET, can exhibit sub-optimal performance on several datasets belonging to different types, such as HSE, DD, ES→MU, and ZINC-Size. In Fig. 3, although SIGNET has a higher median and a compact interquartile range, the top 25% of its performance is still lower than that of some other models. Similar unstable performance can also be found in other competitive methods, such as OCGTL, CVTGAD, and GOOD-D. This finding illustrates the diversity of our benchmark datasets, underscoring the challenge of identifying a "universal method" that performs well across all datasets. This is particularly evident in the AUPRC metric. For instance, on HSE, SIGNET achieves an AUROC of 64.56 vs. 71.42 (SOTA). On DD, it scores 74.53 vs. 80.76 (SOTA). In the inter-dataset shift ES→MU, it achieves 91.43 vs. 99.64 (SOTA), and in the intra-dataset shift ZINC-Size, it scores 57.29 vs. 68.43 (SOTA).

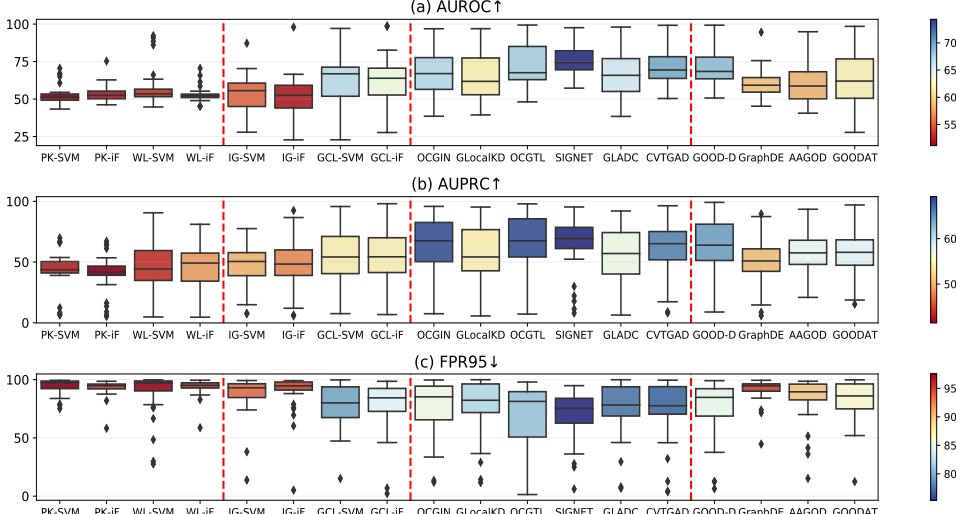

Figure 3: Comparison of the detection performance on 35 datasets in terms of three metrics.

**Observation ❸: Inconsistent performance in terms of different metrics**. From Fig. 3 and Table 8, we can observe that some methods performing well on certain metrics may show unstable performance on other metrics. Specifically, CVTGOD and GOOD-D consistently achieve high AUC values on most datasets, but show more variability in other metrics, particularly in their sub-optimal AUPRC and FPR95 performance. This indicates that methods performing well in overall discrimination (high AUROC) may struggle with precision-recall trade-offs (low AUPRC) and maintaining low false positive rates at high true positive rates (low FPR95). This finding highlights the importance of comprehensive evaluation using multiple metrics.

**Observation ❹: End-to-end methods show consistent superiority over two-step methods.** While two-step methods show notable performance in certain scenarios, end-to-end methods consistently outperform them. Specifically, in Table 3, the average rankings of most end-to-end methods are below 10, while all two-step methods have rankings above 10. This consistent superiority highlights the advantages of integrated and unified learning approaches over segmented and two-step processes.

## 4.2 OOD SENSITIVITY SPECTRUM ON NEAR-OOD AND FAR-OOD (RQ2)

**Experiment design.** To evaluate the OOD sensitivity spectrum of GLAD/GLOD methods in handling near and far OOD samples, we consider two different settings to define near-OOD and far-OOD: **(A) intra-inter dataset setting** and **(B) size-based distance setting**. In setting A, we define the intra-dataset samples with different class labels as the near-OOD, while samples from another dataset are considered far-OOD. In setting B, the size of graphs serves as the measure to divide near and far OOD. We redefine the size of training/testing sets for fair comparison. Please refer to Appendix D.4 for a more detailed description of experimental settings.

**Experimental results.** The performance of GLAD and GLOD methods in distinguishing between ID and near-OOD/far-OOD samples is demonstrated in Fig. 4 and Table 7, where Subfigures (a)-(c) are in setting A and (d) is in setting B. We have the following key observations.

**Observation ❺: Near-OOD samples are harder to detect compared to far-OOD samples.** From Fig. 4, it is evident that end-to-end models generally perform better in detecting far-OOD samples than near-OOD samples. This trend is consistent across various datasets. For instance, in the AIDS dataset, models like GOOD-D, GraphDE, and GOODAT achieve highly better AUROC values for far-OOD conditions than for near-OOD. Similar patterns are observed in the other datasets, where most models display better performance in far-OOD scenarios. This consistent performance discrepancy underscores the challenge of detecting near-OOD samples that closely resemble the ID data, highlighting the need for enhanced model sensitivity to subtle deviations.

**Observation ❻: Poor OOD sensitivity spectrum of several GLAD/GLOD methods in specialized scenarios.** Despite the overall effectiveness of GLAD/GLOD methods, their performance has notable limitations, particularly in specialized scenarios. Firstly, in setting A, GLOD approaches (i.e., GOOD-

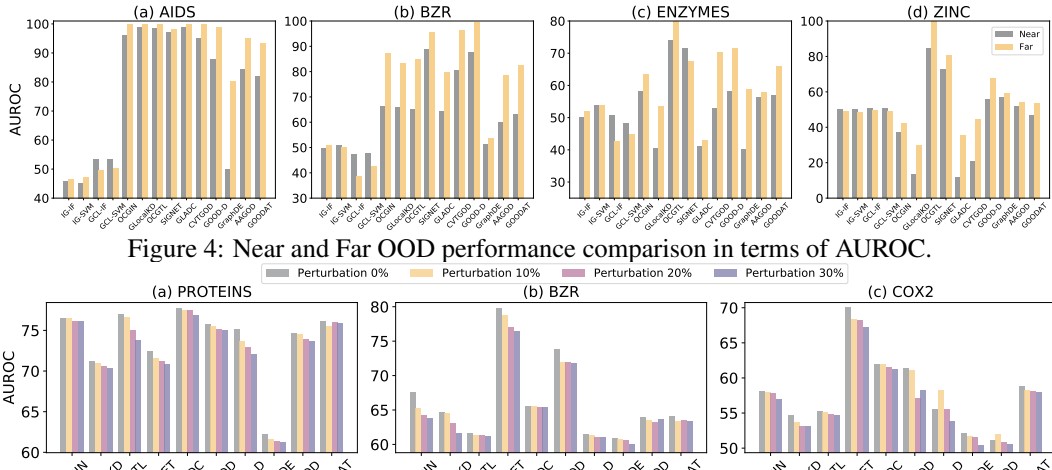

Figure 4: Near and Far OOD performance comparison in terms of AUROC.

Figure 5: Performance of models under different perturbation levels in terms of AUROC.

D and GraphDE) exhibit significant performance gaps between near-OOD and far-OOD conditions. This suggests that when OOD samples are similar to ID samples, the detection capability of GLOD methods is significantly compromised. This limitation is even more pronounced in setting B, where GLAD methods significantly underperform two-step methods. In these scenarios, GLAD methods struggle with size-based deviations, resulting in substantial performance gaps.

## 4.3 ROBUSTNESS UNDER TRAINING SET PERTURBATION (RQ3)

**Experiment design.** In this study, we investigate the robustness of GLAD/GLOD methods against the perturbation of the normal/ID training set by contaminating anomaly/OOD samples. Specifically, we set the perturbation ratios as **0%, 10%, 20%, and 30%** to explore the impacts of different perturbation strengths. For more details, please refer to Appendix D.4.

**Experimental results.** The performance of each GLAD/GLOD method under different strengths of perturbation of the training dataset is demonstrated in Fig. 5, which brings the below observations.

**Observation ❼: Performance degradation with increasing contamination ratio.** From Fig. 5, it is obvious that the performance of GLAD/GLOD methods generally deteriorates as the proportion of OOD samples in the ID training set increases. This trend is evident across multiple datasets and methods. For example, in the PROTEINS, BZR, and COX2 datasets, we observe that models such as GLocalKD, OCGTL, SIGNET, GOOD-D, and others show a consistent decrease in AUROC values as the proportion of OOD contamination increases from 0% to 30%. This decline in performance suggests that the presence of anomaly/OOD samples in the training data introduces noise and confounds the models, making it increasingly difficult for them to distinguish between generalized ID and OOD samples during testing.

**Observation ❽: The sensitivity of different methods/datasets can be diverse.** Although increasing perturbation strength affects most methods, some methods exhibit notable robustness to the perturbation. Specifically, GLADC shows minimal performance degradation across all datasets, while CVTGOD demonstrates impressive stability in the BZR. Conversely, several methods are highly sensitive to perturbations in some cases. For example, OCGTL's performance on the PROTEINS declines sharply as the level of contamination increases. The unstable performance highlights the need for improving the robustness of such methods to ensure reliable performance in real-world scenarios where data contamination is common.

## 4.4 EFFICIENCY ANALYSIS (RQ4)

**Experiment design.** We evaluate the computational efficiency of GLAD/GLOD methods using default hyperparameter settings. Our assessment focuses on two main aspects: **time efficiency** and **memory usage**. See Appendix D.3 for a detailed description.

**Experimental results.** We present the computational efficiency of all methods in terms of time and memory usage in Fig.6(a) and Fig.6(b), respectively. We highlight the below observation.

**Observation ❾: Certain end-to-end methods outperhm two-step methods in terms of both performance and computational costs.** Observation ❹ demonstrates that end-to-end methods usually outperform the two-step methods. In addition to their superior performance, most end-to-end methods exhibit comparable time efficiency and significantly better memory efficiency. As shown in Fig. 6(a), end-to-end methods achieve optimal results much faster

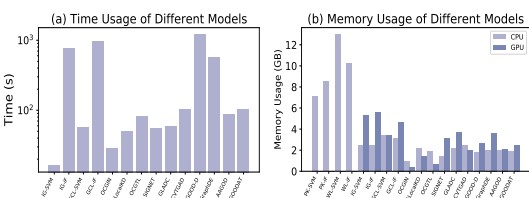

Figure 6: Time and memory usage comparison.

than two-step methods, except for GOOD-D and GraphDE. In contrast, the iF detector in two-step methods significantly increases the time required to reach optimal performance, leading to substantially higher time costs compared to most existing methods. In terms of memory usage, as illustrated in Fig. 6(b), end-to-end methods show much lower CPU and GPU consumption than two-step methods. Although graph kernel methods do not utilize GPU resources, their high CPU consumption is a critical consideration. To sum up, the majority of end-to-end methods may be preferable for GLAD/GLOD tasks due to their advantages in effectiveness and efficiency.

## 5 CONCLUSION AND FUTURE DIRECTIONS

In this paper, we introduce a unified benchmark, UB-GOLD, for graph-level anomaly detection (GLAD) and graph-level out-of-distribution detection (GLOD), which comprehensively compare the performance of 18 GLAD/GLOD methods on 35 datasets across various scenarios and domains. Based on UB-GOLD, we conduct extensive experiments to analyze the effectiveness, OOD sensitivity spectrum, robustness, and efficiency of these methods. Additionally, we provide a visualization and analysis of the Judge Score Distribution in the Appendix F.3. Observation ❶ highlights the intercommunity between GLAD and GLOD tasks. It reveals that many approaches show comparable performance across both tasks, suggesting a significant overlap in the underlying principles and techniques used for detecting anomalies and OOD samples. Several insightful observations are summarized from the results, providing an in-depth understanding of existing methods and inspiration for future work. Our experiments reveal that SOTA methods for GLAD and GLOD exhibit superior performance across both tasks, underscoring the strong interconnection between these two forefront research domains.

Despite the promising results of existing approaches, several critical challenges and research directions remain worthy of future investigation:

- **Universal approaches for diverse datasets.** Observation ❷ suggests the existing GLAD/GLOD methods do not consistently perform well across diverse datasets. In this case, it is a promising opportunity for producing novel approaches that can generally work well on diverse datasets. Such universal approaches should be aware of various structural and attribute characteristics of graph data from diverse domains and be sensitive to different types of OOD and anomaly samples.

- **Awareness of near-OOD samples.** Observations ❺ and ❻ indicate that most existing methods struggle to detect near-OOD samples. Thus, future research is expected to discover more advanced solutions to effectively differentiate between near-OOD and ID samples. Considering the inaccessibility of OOD samples during model training, accurately capturing the patterns of ID data and establishing reliable decision boundaries can be the key to achieving the goal.

- **Robust approaches against unclean training data.** Observations ❼ and ❽ expose that most methods are vulnerable to perturbation (i.e., data contaminated by OOD samples) of training datasets. Considering the difficulty of acquiring clean training data in several real-world applications, future approaches are expected to be more robust against noisy training data.

**Limitation.** Since UB-GOLD mainly focuses on unsupervised GLAD/GLOD tasks, the methods that require anomaly labels (e.g. i-GAD (Zhang et al., 2022)), and domain-specific pre-trained models (e.g., PGR-MOOD (Shen et al., 2024)) are not included for a fair comparison. As a long-term evolving project, UB-GOLD will include these methods in the codebase for quick implementation.

ETHICS STATEMENT

This research is dedicated solely to scientific inquiry, without involving human subjects, animals, or materials that may pose environmental concerns. As such, we do not anticipate any ethical risks or conflicts of interest. We are committed to upholding the highest standards of scientific integrity and ethics to ensure the accuracy and credibility of our findings.

## ACKNOWLEDGMENTS

This work was supported by a grant from the National Natural Science Foundation of China under grants (No.62372211, 62272191), the Foundation of the National Key Research and Development of China (No.2021ZD0112500), the International Science and Technology Cooperation Program of Jilin Province (No.20230402076GH, No. 20240402067GH), and the Science and Technology Development Program of Jilin Province (No. 20220201153GX). Y. Liu and S. Pan were partially supported by Australian Research Council (ARC) under grant DP240101547.

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

## A    WHY UB-GOLD FOCUSES ON UNSUPERVISED SCENARIOS

Node-level anomaly detection (AD) and out-of-distribution (OOD) detection can be performed in both supervised (Tang et al., 2023) and unsupervised (Liu et al., 2022) settings due to their transductive nature. In node-level tasks, the goal is to detect anomalous or OOD nodes within a single graph. Since the entire graph is observed, labeled data is often accessible, and supervised methods can be effectively applied. Supervised approaches leverage these labels to guide the model in identifying anomalies, while unsupervised methods detect deviations from normal node behavior without labels. Both approaches can coexist, as node-level tasks benefit from the complete visibility of the graph during training.

In contrast, graph-level AD/OOD detection poses different challenges, making supervised or semi-supervised approaches less meaningful. Unlike node-level tasks, graph-level detection deals with unseen, independent graphs at test time, where labeled anomaly or OOD samples are rarely available in real-world scenarios. Relying on labeled data in supervised or semi-supervised settings shifts the focus to classification rather than detecting unexpected anomalies. This undermines the core challenge of generalizing across unseen graphs. Furthermore, graph-level tasks share a closer resemblance to computer vision (CV) problems, where the primary challenge is detecting shifts in distribution without labeled anomalies. Our inspiration stems from the success of benchmarks like OpenOOD (Zhang et al., 2023) in CV, where unsupervised methods have proven to be more effective at handling distribution shifts in the absence of labeled OOD data.

By focusing solely on unsupervised methods in UB-GOLD, we ensure that the benchmark reflects the true nature of graph-level AD/OOD detection in real-world applications. Unsupervised approaches eliminate the dependency on labeled data, allowing models to learn from the normal distribution and detect deviations more naturally. This aligns with practical conditions, where labeled OOD or anomaly samples are rarely available. Additionally, an unsupervised framework promotes more robust generalization across diverse datasets, pushing the boundaries of AD/OOD detection in the graph domain, similar to the advancements driven by OpenOOD in computer vision.

## B    DETAILED DESCRIPTION OF DATASETS IN UB-GOLD

The description of datasets in UB-GOLD is given as follows. To ensure the reliability of our benchmark and avoid any data leakage, we carefully curated all dataset splits to guarantee no overlap between ID and OOD samples, or between training and test sets. Each dataset was meticulously prepared to maintain a clear separation, ensuring that OOD samples are entirely distinct from ID data. This strict partitioning is consistently applied across all experimental setups, providing a robust and unbiased evaluation of the methods.

● **Type I: Datasets with intrinsic anomaly.**[2]

Datasets under this type include real anomalies inherently present in the data, defined based on the biological or chemical properties of the samples. These anomalies are characterized by rare or atypical patterns that deviate from the majority of the data. Such intrinsic anomalies are valuable for evaluating the model's ability to detect deviations from the norm, a common challenge in real-world data analysis.

- **HSE (Abdelaziz et al., 2016):** Anomalies in this dataset are samples exhibiting unusual stress response elements that deviate significantly from typical cellular stress responses. These may include rare mutations or outlier stress markers.

- **MMP (Abdelaziz et al., 2016):** Anomalous samples are identified as molecular structures with abnormal expressions of matrix metalloproteinases, often linked to extreme biological behaviors such as uncontrolled cancer metastasis.

- **p53 (Abdelaziz et al., 2016):** The anomalies are instances with abnormal p53 protein behavior or structural deviations, which may indicate dysfunction in apoptosis or tumor suppression mechanisms.

---

[2]Tox21 Challenge Data.

- **PPAR-gamma (Abdelaziz et al., 2016):** Anomalies are molecular samples exhibiting irregular activation or suppression of PPAR-gamma, which is unusual in typical metabolic and inflammatory processes.

● **Type II: Datasets with class-based anomaly.**[3]

This type includes six datasets (PROTEINS, ENZYMES, AIDS, DHFR, BZR, COX2) where the graphs are attributed, meaning that each node contains descriptive features. The remaining datasets are plain graphs without node-level attributes. All datasets with class-based anomalies are derived from graph classification benchmarks, and for anomaly detection tasks, the minority class is treated as anomalies. These datasets are particularly useful for evaluating models' ability to identify rare classes or outliers within a given graph dataset.

- **COLLAB (Yanardag & Vishwanathan, 2015):** Scientific collaboration networks, representing authors and their co-authored papers in various scientific fields.
- **IMDB-BINARY (Yanardag & Vishwanathan, 2015):** Movie collaboration networks, where nodes represent actors and edges denote their co-appearance in films. This dataset helps in analyzing the collaboration patterns within the film industry.
- **REDDIT-BINARY (Yanardag & Vishwanathan, 2015):** Discussion threads from Reddit, each graph representing a thread with nodes as users and edges as interactions. It captures the structure of online discussions and their dynamics.
- **ENZYMES (Schomburg et al., 2004):** Protein tertiary structures, each graph represents an enzyme with nodes as secondary structure elements and edges as spatial adjacencies, crucial for biochemical and functional studies.
- **PROTEINS (Dobson & Doig, 2003):** Comprehensive protein structures and interaction data, where nodes represent amino acids and edges denote interactions, providing insights into protein functions and interactions.
- **DD (Vishwanathan et al., 2010):** Protein-protein interaction networks, capturing the intricate relationships between proteins within biological systems, essential for understanding cellular functions.
- **BZR (Wu et al., 2018):** Benzodiazepine receptor ligands, with nodes representing atoms and edges representing bonds, used for studying molecular interactions with benzodiazepine receptors.
- **AIDS (Morris et al., 2020):** Chemical compounds screened for antiviral activity against HIV, where each graph represents a molecule, useful in drug discovery and antiviral research.
- **COX2 (Morris et al., 2020) :** Cyclooxygenase-2 inhibitors, where nodes and edges represent molecular structures, important for studying anti-inflammatory drug properties.
- **NCI1 (Wale et al., 2008):** Chemical compounds screened for anti-cancer activity, each graph representing a molecule, used for evaluating potential anti-cancer drugs.
- **DHFR (Morris et al., 2020):** Dihydrofolate reductase inhibitors, with graphs representing molecular structures, focusing on compounds inhibiting the DHFR enzyme, vital for cancer and bacterial infection treatments.

● **Type III: Datasets with inter-dataset shift.**

Inter-dataset shift datasets are designed by drawing ID and OOD samples from different but related datasets. This design approach allows us to simulate real-world scenarios where data distributions for both ID and OOD samples are distinct, yet share some underlying similarities.

- **BBBP (Martins et al., 2012):** BBBP is the Blood–brain barrier penetration (BBBP) dataset includes binary labels for over 2000 compounds on their permeability properties.
- **BACE (Subramanian et al., 2016):** BACE provides a series of human $\beta$-secretases as well as their binary label, and all data are experimental values reported in the scientific literature over the last decade.
- **CLINTOX (Gayvert et al., 2016):** The ClinTox dataset compares drugs that have been approved by the FDA with drugs that have failed in clinical trials for toxicity reasons.

---

[3]TUDataset.

- **LIPO (Wu et al., 2018):** The full name of LIPO is Lipophilicity, which is an important feature of drug molecules that affects both membrane permeability and solubility.
- **FREESOLV (Mobley & Guthrie, 2014):** The Free Solvation database (FreeSolv) provides experimental and calculated free energies of hydration of small molecules in water.
- **TOXCAST (Richard et al., 2016):** ToxCast providing toxicology data for a library of compounds based on high-throughput screening and includes qualitative results of over 600 experiments on 8615 compounds.
- **ESOL (Delaney, 2004):** ESOL is a small dataset containing water solubility data for 1128 compounds with the goal of estimating solubility from chemical structures.
- **MUV (Rohrer & Baumann, 2009):** The Maximum Unbiased Validation (MUV) group is another benchmark dataset selected from PubChem BioAssay by applying a modified nearest neighbor analysis
- **TOX21 (Wu et al., 2018):** The dataset contains qualitative toxicity measurements of 8014 compounds against 12 different targets, including nuclear receptors and stress response pathways.
- **SIDER (Altae-Tran et al., 2017):** The Side Effect Resource (SIDER) is a database of marketed drugs and adverse drug reactions which measured for 1427 approved drugs.
- **PTC-MR (Helma et al., 2001):** PTC dataset labels compounds based on their carcinogenicity where MR Indicates that their rodent is a male rat.

● **Type IV: Datasets with intra-dataset shift.**

Intra-dataset shift datasets simulate OOD conditions within a single dataset by defining shifts based on specific attributes such as size, scaffold, or assay. Unlike inter-dataset shifts, which involve drawing ID and OOD samples from different datasets, intra-dataset shifts introduce variations within the same dataset, creating challenges for models that must adapt to these changes without external data sources. Intra-dataset shifts are particularly useful for testing how well a model can handle structural or feature-based changes within a consistent data domain.

- **GOOD (Gui et al., 2022):** A systematic benchmark specifically tailored for the graph OOD problem. We utilize two molecular datasets for OOD detection tasks: (1) GOOD-HIV is a small-scale real-world molecular dataset which aim to predict whether this molecule can inhibit HIV replication. (2) GOOD-ZINC is a real-world molecular property regression dataset from ZINC database and aims at predicting molecular solubility. Each dataset comprises two ID-OOD splitting strategies (scaffold and size), resulting in a total of 4 distinct datasets.
- **DrugOOD (Ji et al., 2023):** This OOD benchmark is designed for AI-aided drug discovery. It includes three ID-OOD splitting strategies: assay, scaffold, and size. These strategies are applied to two measurements (IC50 and EC50), resulting in six datasets. Each dataset comprises a binary classification task aimed at predicting drug target binding affinity.

## C  DETAILED DESCRIPTION OF ALGORITHMS IN UB-GOLD

The description of benchmarking algorithms in UB-GOLD is demonstrated as follows.

● **Graph kernels.**

- **Weisfeiler-Leman Subtree Kernel (WL) (Li et al., 2016) :** This kernel generates graph embeddings by iteratively refining node labels based on subtree patterns. It effectively captures structural similarities within graphs, making it a powerful tool for embedding generation.
- **Propagation Kernel (PK) (Neumann et al., 2016):** This method propagates labels through the graph structure, resulting in embeddings that reflect the graph's topology. It captures relational information within the graph, providing a robust basis for subsequent outlier detection.

● **Self-supervised learning methods.**

- **GraphCL (GCL) (You et al., 2020):** GCL leverages augmentations to learn robust graph-level representations. By contrasting different views of the same graph, the model learns to capture

essential graph structures and properties, making it highly effective for various graph-based tasks. This method is known for its strong performance in unsupervised learning settings.

- **InfoGraph (IG) (Sun et al., 2020):** IG maximizes the mutual information between local and global graph representations to achieve unsupervised and semi-supervised graph-level representation learning. By capturing meaningful information across different levels of the graph, IG effectively learns comprehensive graph embeddings that are useful for a wide range of downstream tasks.

● **Outlier Detectors.**

- **Isolation Forest (iF) (Liu et al., 2008):** This algorithm isolates observations by constructing an ensemble of trees, each built by randomly selecting features and split values. An anomaly score is calculated based on the path length from the root to the leaf node, with shorter paths indicating anomalies.

- **One-Class SVM (OCSVM) (Amer et al., 2013):** OCSVM aims to separate the data from the origin using a hyperplane in a high-dimensional space. Points that lie far from the hyperplane are considered outliers.

● **Graph neural network-based GLAD methods.**

- **OCGIN (Zhao & Akoglu, 2023):** Utilizes a GIN encoder optimized with a Support Vector Data Description (SVDD) objective to identify anomalies within the graph structure. This method employs an end-to-end GNN model with one-class classification for effective anomaly detection.

- **GLocalKD (Ma et al., 2022):** Jointly learns two GNNs and performs graph-level and node-level random knowledge distillation between their learned representations. By leveraging both local and global knowledge, this approach enhances the detection of anomalies.

- **OCGTL (Qiu et al., 2022):** Extends deep one-class classification to a self-supervised detection approach using neural transformations and graph transformation learning as regularization. This technique improves the model's unsupervised anomaly detection capabilities.

- **SIGNET (Liu et al., 2023b):** Proposes a self-interpretable graph-level anomaly detection framework that infers anomaly scores while providing subgraph explanations. By maximizing mutual information of multi-view subgraphs, it achieves both detection and interpretation of anomalies.

- **GLADC (Luo et al., 2022):** Incorporates graph-level adversarial contrastive learning to identify anomalies. Through the creation of adversarial examples and learning robust representations, this method effectively distinguishes between normal and abnormal graphs.

- **CVTGAD (Li et al., 2023):** Introduces a simplified transformer with cross-view attention for unsupervised graph-level anomaly detection. It overcomes the limited receptive field of GNNs by using a transformer-based module to capture relationships between nodes and graphs from both intra-graph and inter-graph perspectives.

● **Graph neural network-based GLOD methods.**

- **GOOD-D (Liu et al., 2023a):** Performs perturbation-free graph data augmentation and utilizes hierarchical contrastive learning on the generated graphs for graph-level OOD detection. By leveraging multiple levels of contrastive learning, GOOD-D enhances the representation learning process, making it robust in distinguishing between in-distribution (ID) and out-of-distribution (OOD) graphs.

- **GraphDE (Li et al., 2022):** Models the generative process of the graph to characterize distribution shifts. Using variational inference, GraphDE infers the environment from which a graph sample is drawn. This generative approach effectively identifies ID and OOD graphs by detecting shifts in the underlying data distribution.

- **AAGOD (Guo et al., 2023):** A data-centric framework for graph OOD detection that operates on well-trained GNNs without retraining. AAGOD uses an Adaptive Amplifier to modify input graphs, highlighting key patterns for OOD detection. Through its Learnable Amplifier Generator (LAG) and Regularized Learning Strategy (RLS), it significantly improves detection performance and efficiency.

- **GOODAT (Wang et al., 2024a):** A test-time graph OOD detection method designed to operate on well-trained GNNs without requiring training data or altering the GNN architecture. GOODAT employs a graph masker and the Graph Information Bottleneck (GIB) principle to extract informative subgraphs. Utilizing GIB-boosted loss functions effectively distinguishes between ID and OOD graphs, achieving strong performance across diverse datasets.

# D  SUPPLEMENTAL INFORMATION OF UB-GOLD

## D.1  DEFINITION OF GLAD AND GLOD

**GLAD definition.** For GLAD, the task is to detect anomalous graphs within a given distribution $\mathbb{P}^{in}$. The objective is to assign anomaly scores $S(G)$ such that anomalous graphs are distinguishable from normal graphs based on a threshold $\tau$:

$$g(G; \tau, S) = \begin{cases} 0 \text{ (Anomalous)}, & \text{if } S(G) \leq \tau, \\ 1 \text{ (Normal)}, & \text{if } S(G) > \tau. \end{cases} \quad (3)$$

The theoretical objective for the anomaly scoring function $S(G)$ is to maximize the separation between the distributions of normal and anomalous graphs:

$$\max_{S} \mathbb{E}_{G \sim \mathbb{P}^{\text{normal}}} S(G) - \mathbb{E}_{G \sim \mathbb{P}^{\text{anomalous}}} S(G), \quad (4)$$

where $\mathbb{P}^{\text{normal}}$ and $\mathbb{P}^{\text{anomalous}}$ represent the idealized distributions of normal and anomalous graphs. While this objective cannot be directly optimized during training due to the lack of access to test distributions, it provides a guiding principle for designing $S(G)$. Practical approaches typically rely on approximations or surrogate objectives based on training data, such as unsupervised or self-supervised learning frameworks, to approximate the separation of score distributions. When the separation is achieved, a threshold $\tau$ can effectively classify graphs as normal (1) or anomalous (0), aligning with the GLOD framework.

**GLOD definition.** For GLOD, the task is to detect graphs in a test dataset that originate from a different distribution $\mathbb{P}^{out}$ compared to the training distribution $\mathbb{P}^{in}$. The goal is to assign OOD scores $J(G)$ such that OOD graphs have significantly lower scores than ID graphs, enabling a threshold-based classification.

$$g(G; \tau, J) = \begin{cases} 0 \text{ (OOD)}, & \text{if } J(G) \leq \tau, \\ 1 \text{ (ID)}, & \text{if } J(G) > \tau. \end{cases} \quad (5)$$

The theoretical objective for the OOD scoring function $J(G)$ is to maximize the separation between the distributions of ID and OOD graphs:

$$\max_{J} \mathbb{E}_{G \sim \mathbb{P}^{\text{in}}} J(G) - \mathbb{E}_{G \sim \mathbb{P}^{\text{out}}} J(G), \quad (6)$$

where $\mathbb{P}^{\text{in}}$ and $\mathbb{P}^{\text{out}}$ represent the idealized distributions of ID and OOD graphs, respectively. Although these distributions are not directly accessible during training (since OOD samples are not present in the training phase), they provide a theoretical goal for designing $J(G)$. In practice, approximations are used based on the available training data to model the OOD detection function. When the distributions of ID and OOD scores are well-separated, a threshold $\tau$ can effectively classify graphs as ID (1) or OOD (0).

## D.2  METRICS

In UB-GOLD, we consider three metrics for evaluation. Their definitions are given as follows.

- **AUROC:** Fundamental for both GLOD and GLAD, AUROC measures a model's ability to distinguish between normal and anomalous or OOD instances across various threshold levels. A higher AUROC value indicates better performance in correctly classifying positives (anomalies or OOD instances) and negatives (normal instances), making it crucial for evaluating the overall effectiveness of detection algorithms.

Table 4: Hyper-parameter search space of all implemented methods.

| Algorithm | Hyper-parameter | Search Space |
|---|---|---|
| **General Settings** | hidden size | 16, 32, 64, 128 |
| | dropout | 0, 0.1, 0.2, 0.3 |
| | layers | 1, 2, 3, 4 |
| | learning rate | 1e-1, 1e-2, 1e-3, 1e-4 |
| **GOOD-D** | str_dim | 8, 16, 24, 32 |
| | cluster number | 2, 3, 4 |
| | $\alpha$ | [0, 1.0] |
| **GraphDE** | dropedge | 0, 0.1, 0.2, 0.3 |
| | dropnode | 0, 0.1, 0.2, 0.3 |
| | model type | graphde-v, graphde-a |
| **CVTGAD** | str_dim | 8, 16, 24, 32 |
| | cluster number | 2, 3, 4 |
| | $\alpha$ | [0, 1.0] |
| | pooling | mean, max |
| **GLADC** | hidden size | 32, 64, 128, 256 |
| | output size | 32, 64, 128 |
| **GLocalKD** | clip | 0.10, 0.15, 0.20 |
| | nobn | True, False |
| | nobias | True, False |
| **OCGTL** | hidden size | 32, 64, 128 |
| | layers | 2, 3, 4, 5 |
| **SIGNET** | pooling | add, max |
| | readout | concat, add, last |
| | layers | 3, 4, 5 |
| **OCGIN** | aggregation | mean, add, max |
| | bias | True, False |
| **AAGOD** | $\lambda$ | 10, 50, 100 |
| **GOODAT** | $\alpha$ | 0.1, 0.3, 0.5, 0.7, 0.9 |
| | $\beta$ | 0.03, 0.05, 0.07 |
| **GCL+kernel** | tree number | 200, 250, 300 |
| | sample ratio | 0.3, 0.4, 0.5, 0.6 |
| **KernelGLAD** | tree number | 200, 250, 300 |
| | sample ratio | 0.3, 0.4, 0.5, 0.6 |
| | neighbors | 20, 30, 40 |
| | leaves | 25, 30, 35 |
| | WL iteration | 3, 4, 5, 6, 7 |

- **AUPRC:** Gains importance in imbalanced datasets, common in AD where anomalies are rare, and in GLOD where OOD instances are infrequent. This metric focuses on precision (the accuracy of positive predictions) and recall (the model's ability to detect all positive cases), providing a clear measure of performance in scenarios where positive cases are critical and more challenging to detect.

- **FPR95:** Evaluates the number of false positives accepted when the model correctly identifies 95% of true positives. This metric is particularly useful in settings where missing an anomaly or an OOD instance can lead to significant consequences, emphasizing the need for models that maintain high sensitivity without sacrificing specificity.

Table 5: Dataset statistics including ID Train, ID Test, Near OOD, and Far OOD counts. Additionally, Unseen Class (UC), Unseen Dataset (UD), Near Size (NS), and Far Size (FS).

| Scenario Type | Data | ID Train | ID/Near/Far Test | Near OOD | Far OOD |
|---|---|---|---|---|---|
| Class-based Anomaly | AIDS | 1280 | 80 | AIDS(UC) | DHFR(UD) |
| | BZR | 69 | 17 | BZR(UC) | COX2(UD) |
| | ENZYMES | 400 | 20 | ENZYMES(UC) | PROTEINS(UD) |
| Size-based | GOOD-ZINC-Size | 1000 | 500 | ZINC(NS) | ZINC(FS) |

Table 6: Dataset statistics including ID Train, ID Train with different perturbation levels, ID Test, and OOD Test counts.

| Data | ID Train | ID Train (10%) | ID Train (20%) | ID Train (30%) | ID Test | OOD Test |
|---|---|---|---|---|---|---|
| BZR | 69 | 76 | 82 | 89 | 17 | 44 |
| PROTEINS | 360 | 374 | 387 | 400 | 90 | 93 |
| COX2 | 81 | 89 | 96 | 103 | 21 | 51 |

## D.3 ADDITIONAL EXPERIMENTAL DETAILS

**Implementation Details.** To ensure a comprehensive evaluation and maintain fairness across a broad spectrum of models, we develop an open-source toolkit named UB-GOLD. This toolkit is built on top of Pytorch 2.01 (Paszke et al., 2019), torch_geometric 2.4.0 (Fey & Lenssen, 2019) and DGL 2.1.0 (Wang et al., 2019). We implement graph kernel methods with the DGL library. All other models are unified using the torch_geometric library. GCL and IG are included via the PYGCL library (Zhu et al., 2021).

**Hardware Specifications.** All our experiments were carried out on a Linux server with an Intel(R) Xeon(R) Gold 5120 2.20GHz CPU, 160GB RAM, and NVIDIA A40 GPU, 48GB RAM.

**Hyperparameter Settings.** Table 4 provides a comprehensive list of all hyperparameters used in our random search complete with their search spaces. For the design of the default hyperparameters please refer to our code base in. /benchmark/Source.

**Efficiency Analysis (Sec. 4.4).** We evaluate the computational efficiency of GLOD and GLAD methods using default hyperparameter settings. Our assessment focuses on two main aspects:

- **Time Efficiency**: We record the average time each method takes to achieve the best results across all datasets, providing insights into their processing speed.

- **Resource Usage**: We monitor each method's CPU and GPU consumption (on COLLAB) during experiments, determining their demand for computational resources.

This setup allows us to measure the methods' efficiency directly and reliably, reflecting their practicality for real-world application.

## D.4 NEW EXPERIMENTS DATASET SPLIT

In the experiments for OOD sensitivity spectrum and robustness, we do not follow the original split but utilize experiment-specific splits for fair comparison. The details are as follows.

**OOD sensitivity spectrum on Near-OOD and Far-OOD (Sec. 4.2).** To rigorously evaluate the performance of GLOD and GLAD methods in handling near and far OOD conditions, we have implemented a distinct partitioning strategy for this research question. Unlike the setup in RQ1, here we ensure that the number of samples in the Near OOD, Far OOD, and ID Test groups are precisely equal, detailed in Table 5. This balanced configuration is designed to provide a fair comparison across different degrees of OOD scenarios, and it includes two specific setups:

- **Intra-inter dataset setting:** We utilize the class-based anomaly partitioning method to set the ID Train and ID Test, along with the Near OOD Test. The Far OOD Test employs datasets from the Inter-Dataset Shift category, representing more significant deviations.

- **Size-based distance setting:** We maintain the same ID Train and ID Test groupings of Intra-Dataset Shift. However, for the Near OOD and Far OOD tests, we categorize the samples based on differing graph sizes, with smaller sizes representing Near OOD and larger sizes for Far OOD.

These settings are designed to rigorously test the capability of GLOD and GLAD methods to recognize and differentiate between subtle and substantial distribution shifts, thereby assessing their effectiveness in realistic and challenging environments.

**Robustness under Training Set Perturbation (Sec. 4.3).** In this study, we investigate the robustness of GLOD and GLAD methods against the contamination of the ID training set with OOD samples. In Table 6, we begin by partitioning the OOD test dataset, randomly selecting 30% of its samples to be mixed into the ID training set. The remaining 70% of the OOD test dataset is kept intact for performance evaluation. This procedure is repeated to create four distinct experimental groups where 0%, 10%, 20%, and 30% of the originally selected OOD samples are added to the ID training dataset. These modifications allow us to systematically explore how progressively increasing the proportion of OOD samples within the ID training set affects the methods' ability to identify and differentiate true OOD instances during testing accurately.

## E    REPRODUCIBILITY

Ensuring the reproducibility of experimental results is a core principle of UB-GOLD. Below, we outline the measures we have taken to achieve this:

**Accessibility.** All datasets, algorithm implementations, and experimental configurations are freely accessible through our open-source project at `https://github.com/UB-GOLD/UB-GOLD`. No special requests or permissions are needed to access the resources.

**Datasets.** Our datasets are publicly available and include TUDataset, OGB, TOX21, DrugOOD, and GOOD. Among them, TUDataset (Morris et al., 2020), OGB (Hu et al., 2020), and TOX21 (Abdelaziz et al., 2016) are licensed under the MIT License. DrugOOD (Ji et al., 2023) is licensed under the GNU General Public License 3.0. GOOD (Gui et al., 2022) is licensed under GPL-3.0.

All these datasets are permitted by their authors for academic use and contain no personally identifiable information or offensive content.

**Documentation and Usage.** We provide comprehensive documentation to facilitate easy use of our library. The code includes thorough comments to enhance readability. Users can reproduce experimental results by following the examples, which outline how to run the code with specific data, methods, and GPU configuration arguments.

**License.** UB-GOLD is distributed under the MIT license, ensuring wide usability and adaptability.

**Code Maintenance.** We are dedicated to regularly updating our codebase, addressing user feedback, and incorporating community contributions. We also enforce strict version control to maintain reproducibility throughout the code maintenance process.

With these measures, UB-GOLD aims to foster transparency, accessibility, and collaboration within the research community.

## F    ADDITIONAL EXPERIMENTAL RESULTS AND ANALYSES

### F.1    ADDITIONAL EXPERIMENTAL RESULTS

In this section, we present additional experimental results, providing comprehensive insights into the performance of our models across different datasets and metrics.

**Main experiments.** Table 8 presents the complete results of the main experiment. Each model is executed 5 times with varying random seeds, and the mean scores along with standard deviations are reported. "Avg. AUROC", "Avg. FPR95", "Avg. AUPRC", and "Avg. Rank" indicate the average AUROC, FPR95, AUPRC, and rank across all datasets, respectively. We observe that the End-to-End approach stands out in AUROC, achieving the best overall results. However, in the metrics of AUPRC and FPR95, SSL-D in the two-step approach also performs well, showing competitive results.

Table 7: Near and Far OOD performance comparison.

| Model | Type | AIDS | | | BZR | | | ENZYMES | | | ZINC-Size | | |
|---|---|---|---|---|---|---|---|---|---|---|---|---|---|
| | | AUROC ↑ | AUPR ↑ | FPR95 ↓ | AUROC ↑ | AUPR ↑ | FPR95 ↓ | AUROC ↑ | AUPR ↑ | FPR95 ↓ | AUROC ↑ | AUPR ↑ | FPR95 ↓ |
| **IG-iF** | Far OOD | 46.50 | 48.88 | 93.75 | 50.80 | 53.88 | 94.12 | 51.87 | 56.31 | 92.00 | 48.95 | 50.19 | 96.00 |
| | Near OOD | 45.92 | 47.75 | 93.75 | 49.58 | 54.43 | 91.76 | 50.15 | 54.76 | 89.00 | 50.39 | 50.90 | 94.72 |
| **IG-SVM** | Far OOD | 47.17 | 48.57 | 95.75 | 50.17 | 51.98 | 95.29 | 53.88 | 55.27 | 85.00 | 48.66 | 50.02 | 95.64 |
| | Near OOD | 45.01 | 46.42 | 95.50 | 50.83 | 55.36 | 96.47 | 53.85 | 55.51 | 85.00 | 50.14 | 50.46 | 94.92 |
| **GCL-iF** | Far OOD | 49.73 | 50.60 | 94.00 | 38.55 | 46.59 | 100.00 | 42.85 | 50.73 | 93.00 | 48.95 | 49.71 | 95.32 |
| | Near OOD | 53.53 | 53.86 | 94.50 | 47.23 | 54.29 | 98.82 | 50.65 | 52.68 | 88.00 | 50.51 | 50.00 | 94.00 |
| **GCL-SVM** | Far OOD | 50.26 | 50.60 | 93.00 | 42.56 | 49.85 | 97.65 | 44.82 | 54.64 | 91.00 | 49.43 | 49.90 | 95.20 |
| | Near OOD | 53.52 | 53.41 | 93.50 | 47.47 | 51.52 | 95.29 | 48.20 | 53.17 | 90.00 | 50.55 | 50.36 | 93.76 |
| **OCGIN** | Far OOD | 99.88 | 99.88 | 0.75 | 87.20 | 83.44 | 37.65 | 63.45 | 72.12 | 93.00 | 41.97 | 45.39 | 97.00 |
| | Near OOD | 96.01 | 95.95 | 13.25 | 66.16 | 59.78 | 50.59 | 58.35 | 63.39 | 89.00 | 36.96 | 45.98 | 99.56 |
| **GLocalKD** | Far OOD | 100.00 | 100.00 | 0.00 | 83.39 | 75.63 | 35.29 | 53.70 | 67.07 | 85.00 | 30.07 | 34.50 | 95.40 |
| | Near OOD | 99.89 | 99.90 | 0.00 | 65.81 | 58.54 | 51.76 | 40.45 | 54.30 | 85.00 | 13.14 | 31.09 | 99.92 |
| **OCGTL** | Far OOD | 100.00 | 100.00 | 0.00 | 84.64 | 77.25 | 28.24 | 80.10 | 77.04 | 41.00 | 99.58 | 99.32 | 1.00 |
| | Near OOD | 99.61 | 99.66 | 0.00 | 65.19 | 58.08 | 48.24 | 74.20 | 69.04 | 51.00 | 84.40 | 70.09 | 23.44 |
| **SIGNET** | Far OOD | 98.00 | 93.07 | 2.00 | 95.50 | 90.38 | 11.76 | 67.48 | 67.05 | 86.00 | 80.48 | 79.23 | 66.96 |
| | Near OOD | 97.16 | 91.02 | 2.50 | 88.93 | 83.93 | 85.88 | 71.60 | 71.27 | 68.00 | 72.90 | 71.41 | 82.64 |
| **GLADC** | Far OOD | 100.00 | 100.00 | 0.00 | 79.58 | 69.52 | 41.18 | 43.00 | 57.80 | 90.00 | 35.55 | 40.40 | 98.12 |
| | Near OOD | 98.74 | 98.60 | 3.75 | 64.36 | 56.84 | 52.94 | 41.25 | 43.32 | 85.00 | 11.63 | 32.34 | 100.00 |
| **CVTGOD** | Far OOD | 99.94 | 99.94 | 0.25 | 93.91 | 91.30 | 17.65 | 70.20 | 76.29 | 87.00 | 44.57 | 51.56 | 95.44 |
| | Near OOD | 94.93 | 95.01 | 24.00 | 80.48 | 72.37 | 63.53 | 53.05 | 58.40 | 88.00 | 20.68 | 38.52 | 99.84 |
| **GOOD-D** | Far OOD | 98.95 | 98.93 | 5.75 | 99.31 | 99.37 | 9.41 | 71.60 | 77.06 | 83.00 | 67.73 | 67.68 | 88.00 |
| | Near OOD | 87.80 | 88.60 | 43.25 | 87.54 | 86.49 | 69.41 | 58.10 | 62.40 | 88.00 | 55.85 | 56.94 | 94.36 |
| **GraphDE** | Far OOD | 80.25 | 90.13 | 100.00 | 51.18 | 50.91 | 96.47 | 58.75 | 64.84 | 100.00 | 59.01 | 65.91 | 98.80 |
| | Near OOD | 50.09 | 71.87 | 100.00 | 49.41 | 50.61 | 97.65 | 40.30 | 47.18 | 100.00 | 56.61 | 57.25 | 98.00 |
| **AAGOD** | Far OOD | 94.95 | 94.80 | 8.25 | 78.39 | 75.42 | 21.50 | 57.92 | 64.35 | 87.00 | 54.23 | 60.45 | 94.00 |
| | Near OOD | 84.32 | 83.45 | 18.75 | 60.08 | 57.80 | 42.34 | 56.34 | 58.72 | 88.00 | 51.60 | 55.23 | 96.00 |
| **GOODAT** | Far OOD | 93.28 | 92.88 | 10.00 | 82.43 | 80.32 | 24.11 | 65.97 | 71.25 | 89.00 | 53.78 | 60.25 | 94.50 |
| | Near OOD | 81.93 | 80.75 | 22.50 | 62.92 | 61.23 | 45.36 | 56.98 | 60.12 | 88.00 | 46.77 | 52.03 | 95.00 |

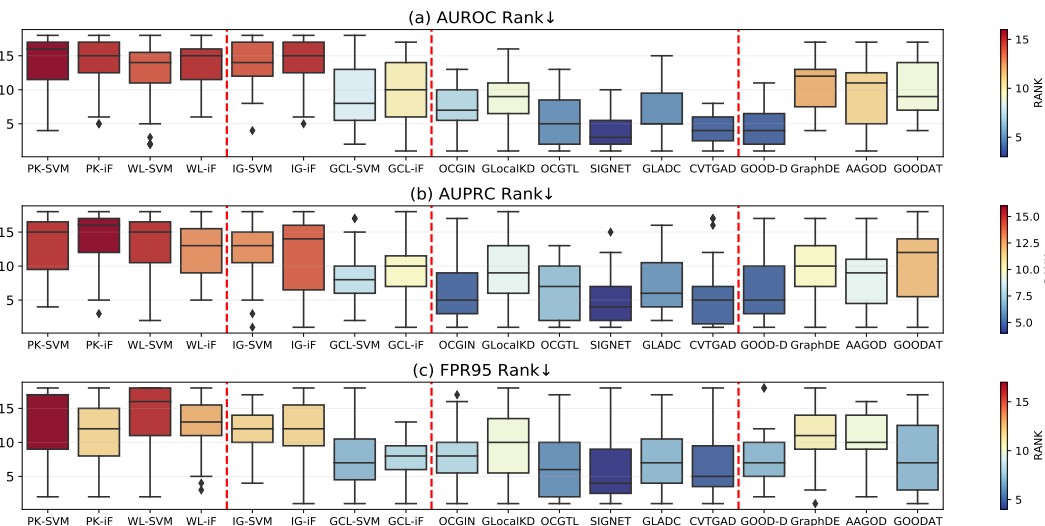

Figure 7: Comparison of the ranking on 35 datasets in terms of three metrics.

**Ranking experiments.** Fig. 7 shows the ranking of models across 35 datasets in terms of their performance. It provides a visual representation of the comparative ranking, allowing for an easy assessment of how different models rank against each other across a large number of datasets.

**Near and far OOD performance.** We provide the full results of Near-OOD and Far-OOD evaluations for three metrics across four datasets, as shown in Table 7. This table allows for a detailed comparison of Near and Far OOD performance, showcasing how our models perform under different out-of-distribution scenarios. Further confirming our findings in Observations❺ and ❻.

**Robustness under Training Set Perturbation.** While Observations ❼ and ❽ are based on AUROC, we also provide AUPRC and FPR95 results in this section. However, it is important to note that since AUPRC and FPR95 are computed based on the model's performance on AUROC, the conclusions drawn from AUROC do not necessarily extend to these other metrics. As shown in Fig.9 and Fig.10,

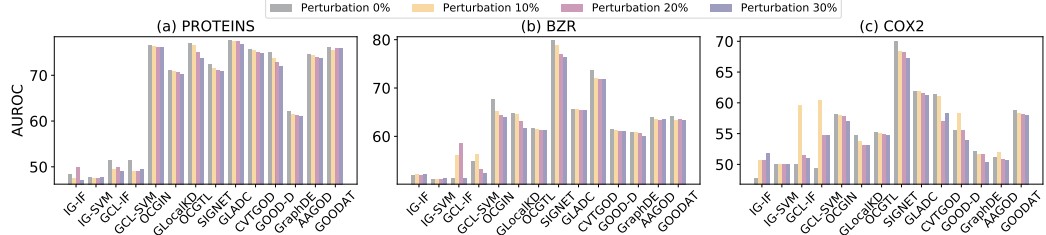

Figure 8: Performance of models under different perturbation levels in terms of AUROC.

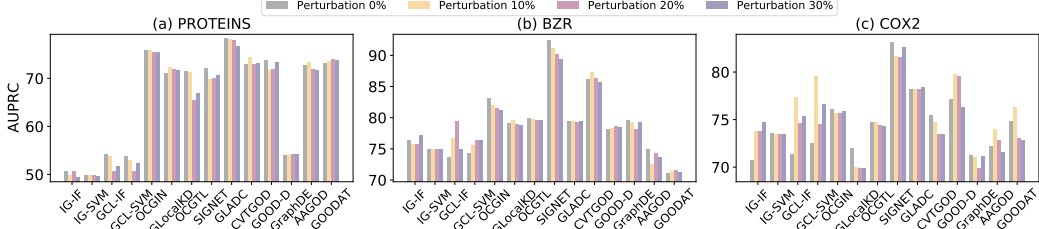

Figure 9: Performance of models under different perturbation levels in terms of AUPRC.

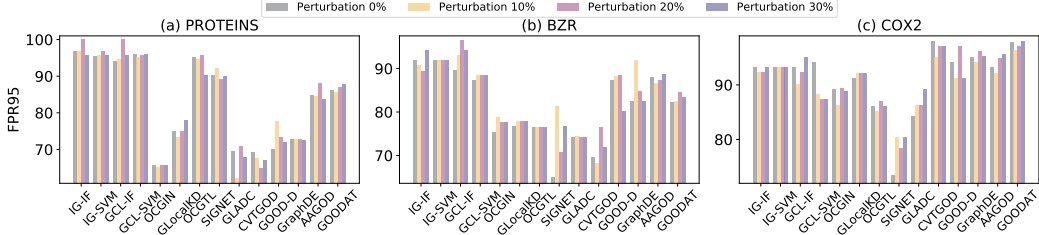

Figure 10: Performance of models under different perturbation levels in terms of FPR95.

the results for AUPRC and FPR95 do not exhibit the same consistent trends, indicating that the behavior of models may vary across different evaluation metrics.

## F.2 ADDITIONAL EXPERIMENTAL ANALYSES

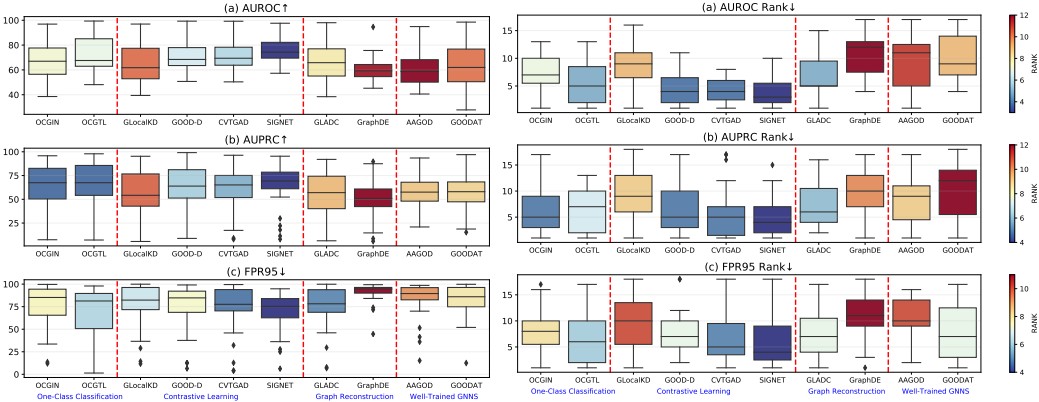

Figure 11: Scores and ranks for four types of methods.

In this section, as illustrated in Fig. 2, we categorize the GLAD/GLOD methods into four main technical groups:

- **One-Class Classification**: OCGIN, OCGTL.
- **Contrastive Learning**: GLocalKD, GOOD-D, CVTGOD, SIGNET.
- **Graph Reconstruction**: GLADC, GraphDE.

- **Well-Trained GNNs**: AAGOD, GOODAT.

From Fig.11, we observe that contrastive learning methods generally perform better and exhibit more stable results compared to other methods. This performance difference can be attributed to several key factors:

**Rich Representations in Contrastive Learning**: Contrastive learning methods (GLocalKD, GOOD-D, CVTGOD, SIGNET) demonstrate superior and stable performance across all metrics, particularly AUROC and AUPRC. This strong performance can be attributed to their ability to maximize mutual information between different augmentations of the same graph (positive pairs) and minimize it for negative pairs. By capturing and preserving the most relevant and rich graph features, these methods produce robust, discriminative representations that generalize well in OOD detection. Their compact interquartile ranges indicate stable performance across datasets.

**Sensitivity to Anomalies in One-Class Classification**: One-class classification methods (OCGIN, OCGTL) perform well overall, particularly in FPR95. These methods excel by learning a dense, representative boundary around the in-distribution (ID) data, treating everything outside this boundary as OOD. Their strength lies in handling compact, well-defined distributions, making them particularly effective in scenarios where OOD samples deviate significantly from the ID distribution. This is why they show relatively consistent performance across datasets. However, when the OOD samples closely resemble the ID samples, the separation becomes more challenging, leading to some variability in performance on metrics like AUROC and AUPRC.

**Limitations of Graph Reconstruction**: Graph reconstruction methods (GLADC, GraphDE) show more variability. GLADC performs reasonably well on AUROC but struggles with AUPRC and FPR95. These methods reconstruct the input graph and detect anomalies based on reconstruction errors, but when the reconstruction error does not strongly correlate with OOD instances, their effectiveness decreases. This is especially true for GraphDE, which shows poor performance and significant variability.

**Dependency on Pre-trained Models in Well-Trained GNNs**: Well-trained GNNs (AAGOD, GOODAT) show mixed results, depending on the availability of pre-trained model parameters. These methods rely heavily on the pre-trained GNNs' quality. GOODAT is more stable across datasets, while AAGOD displays higher variability. The methods perform well when high-quality pre-trained parameters are available, but retraining without these parameters leads to a noticeable performance drop, indicating a strong dependency on the initial pre-training phase.

Overall, contrastive learning methods stand out for their ability to capture high mutual information between graph augmentations, resulting in robust and generalizable representations. One-class classification methods perform well, especially in handling distinct OOD cases but show some sensitivity when OOD samples closely resemble ID data. Graph reconstruction methods face challenges in correlating reconstruction errors with OOD detection, leading to inconsistent results. Well-trained GNNs can be effective but are highly dependent on pre-trained models, impacting their consistency across different scenarios. This analysis highlights the importance of selecting methods that align with the characteristics of the task and data.

### F.3 JUDGE SCORE DISTRIBUTION ANALYSIS

As illustrated in Fig.12, we visualize the OOD Judge score distributions for eight baseline models across three datasets (AIDS, BZR, Tox21_HSE, and AI-DH). The X-axis shows the OOD Judge score, and the Y-axis shows the frequency. These plots demonstrate how OOD and ID samples are distributed based on their scores, highlighting the separation between these distributions. This analysis directly assesses the models' ability to distinguish OOD from ID samples, which is more task-relevant than visualizing the embedding space.

Most methods perform well on the AIDS dataset, with clear separation between OOD (red) and ID (blue) distributions. However, methods like GraphDE and SIGENT exhibit noticeable differences in distribution shapes, suggesting variability in decision boundaries and sensitivity to data characteristics.

In contrast, all methods show poor performance on the Tox21_HSE dataset, with overlapping OOD and ID distributions. This overlap indicates a failure to effectively distinguish OOD from ID samples, suggesting that the dataset presents significant challenges for OOD detection.

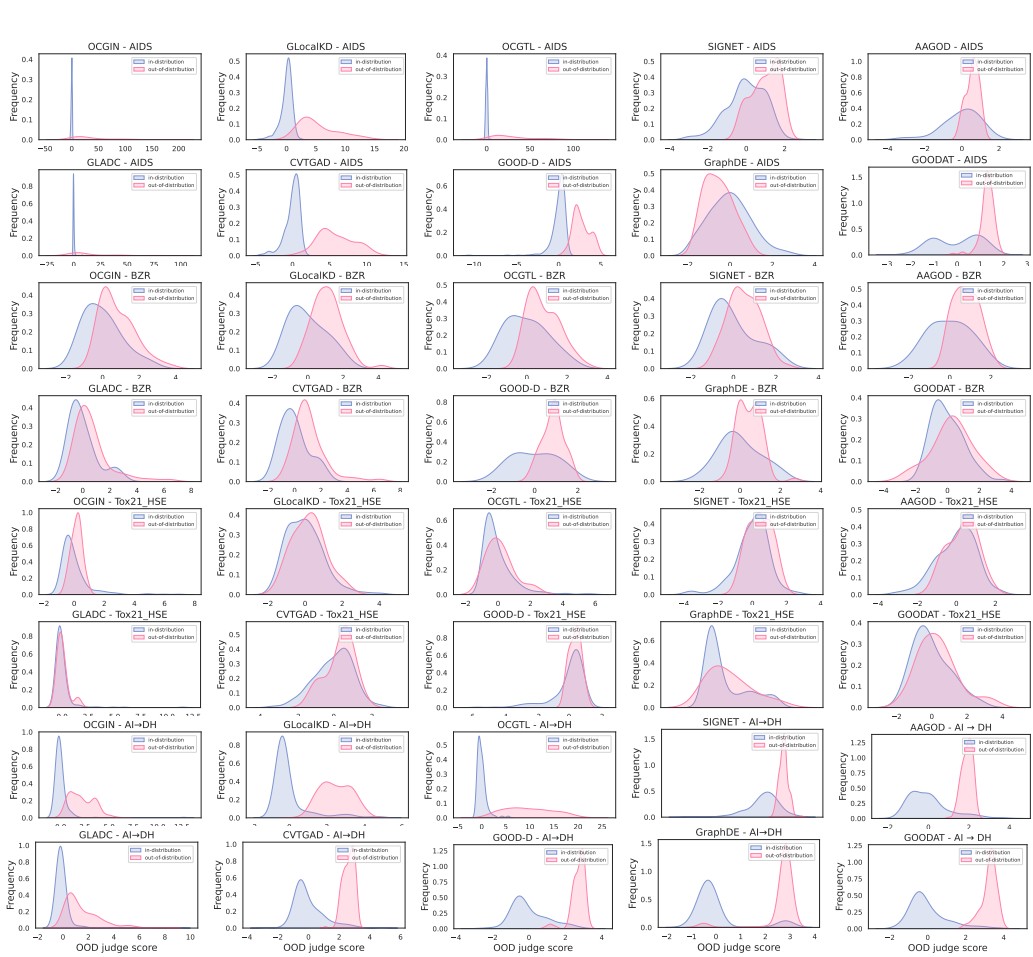

Figure 12: OOD Judge score distributions.

Table 8: Comparison in terms of AUPRC (top), AUROC (middle), and FPR95 (bottom). The best three results are highlighted using 1st , 2nd , and 3rd . Avg. AUROC", Avg. FPR95", "Avg. AUPRC" and "Avg. Rank" indicate the average AUROC, FPR95, AUPRC, and rank across all datasets.

**AUPRC (top)**

| | GK-D (two-step) | | | | SSL-D (two-step) | | | | OCGIN | GLocalKD | OCGTL | SIGNET | GLADC | CVTGAD | GOOD-D | GraphDE | AAGOD | GOODAT |
|---|---|---|---|---|---|---|---|---|---|---|---|---|---|---|---|---|---|---|
| | PK-SVM | PK-iF | WL-SVM | WL-iF | IG-SVM | IG-iF | GCL-SVM | GCL-iF | | | | | | | | | | |
| p53 | 49.17±0.46 | 54.05±0.25 | 57.69±0.57 | 54.40±0.41 | 68.11±0.07 | 60.85±0.68 | 68.61±0.14 | 64.60±0.30 | 68.35±1.20 | 65.43±0.29 | 67.58±0.50 | 68.10±0.15 | 65.82±0.21 | 69.40±0.24 | 67.82±0.37 | 62.59±0.56 | 50.13±0.67 | 61.71±0.35 |
| HSE | 60.72±0.49 | 56.49±0.66 | 63.27±0.26 | 52.98±0.53 | 60.33±0.59 | 22.77±0.63 | 67.40±0.16 | 63.95±0.39 | 71.42±0.10 | 60.21±0.12 | 63.36±0.69 | 64.56±0.60 | 61.37±0.48 | 70.52±0.30 | 68.71±0.22 | 62.47±0.27 | 54.60±0.46 | 61.99±0.80 |
| MMP | 51.03±0.44 | 49.95±1.38 | 55.50±0.67 | 51.98±0.39 | 57.72±0.58 | 52.58±0.29 | 69.91±0.13 | 71.31±0.41 | 69.37±0.12 | 68.12±0.23 | 67.51±0.27 | 71.23±0.18 | 70.03±0.46 | 70.58±0.22 | 71.41±0.51 | 60.12±0.34 | 62.92±0.78 | 67.49±0.18 |
| PPAR | 53.74±0.21 | 48.42±0.16 | 57.76±0.33 | 49.45±1.12 | 61.78±0.66 | 63.22±0.22 | 68.37±0.44 | 69.88±0.28 | 67.75±0.34 | 65.29±0.58 | 66.43±0.67 | 68.88±0.42 | 69.43±0.36 | 68.83±0.10 | 68.24±0.19 | 66.31±0.30 | 57.90±0.56 | 66.95±0.52 |
| COLLAB | 49.72±0.60 | 51.38±0.20 | 54.62±1.28 | 51.41±0.09 | 36.47±0.42 | 38.18±0.34 | 44.91±0.56 | 45.44±0.83 | 60.58±0.27 | 51.85±0.18 | 48.13±0.41 | 72.45±0.11 | 54.32±0.37 | 71.01±0.58 | 69.34±0.55 | 46.77±0.48 | 50.14±0.35 | 44.91±0.19 |
| IMDB-B | 51.75±0.30 | 52.83±0.51 | 52.98±0.69 | 51.79±0.32 | 30.89±0.48 | 45.64±0.67 | 68.00±0.22 | 63.88±0.15 | 61.47±0.18 | 53.31±0.53 | 55.27±0.24 | 70.12±0.61 | 69.82±0.13 | 66.68±0.14 | 59.25±0.39 | 58.43±0.22 | 65.36±0.84 | |
| REDDIT-B | 48.36±0.67 | 46.19±0.21 | 49.18±0.50 | 49.84±0.11 | 60.33±0.32 | 52.51±0.62 | 84.49±0.20 | 82.64±0.42 | 82.10±0.37 | 80.32±0.16 | 89.92±0.11 | 85.24±0.45 | 78.87±0.56 | 87.43±0.60 | 89.43±0.22 | 63.42±0.30 | 68.78±0.23 | 80.31±0.77 |
| ENZYMES | 52.45±0.29 | 49.82±0.67 | 53.75±0.51 | 51.03±0.42 | 60.97±0.19 | 53.94±0.66 | 62.73±0.21 | 63.09±0.26 | 62.44±0.38 | 61.75±0.10 | 63.59±0.11 | 63.12±0.52 | 63.44±0.30 | 68.56±0.43 | 64.58±0.57 | 52.10±0.65 | 58.70±0.35 | 52.33±0.59 |
| PROTEINS | 49.43±0.69 | 61.24±0.34 | 53.85±0.26 | 65.75±0.35 | 61.15±0.11 | 52.78±0.58 | 72.61±0.64 | 72.60±0.20 | 76.46±0.13 | 77.29±0.41 | 72.89±0.57 | 75.86±0.30 | 77.43±0.19 | 76.49±0.29 | 76.02±0.17 | 68.81±0.33 | 75.04±0.25 | 77.92±0.75 |
| DD | 47.69±0.24 | 75.29±0.46 | 47.98±0.32 | 70.49±0.28 | 70.33±0.13 | 42.67±0.59 | 76.43±0.45 | 65.41±0.60 | 79.08±0.19 | 80.76±0.50 | 77.76±0.48 | 74.53±0.11 | 76.54±0.25 | 78.84±0.40 | 79.91±0.65 | 60.49±0.17 | 74.00±0.21 | 77.62±0.26 |
| BZR | 46.67±0.52 | 59.05±0.29 | 51.16±0.36 | 51.71±0.45 | 41.50±0.24 | 45.42±0.34 | 68.93±0.63 | 67.81±0.32 | 69.13±0.13 | 68.55±0.15 | 51.89±0.46 | 80.79±0.58 | 68.23±0.31 | 77.69±0.28 | 73.28±0.40 | 65.94±0.12 | 64.52±0.77 | 64.77±0.12 |
| AIDS | 50.93±0.19 | 52.01±0.53 | 52.56±0.41 | 61.42±0.50 | 87.20±0.18 | 97.96±0.36 | 95.43±0.65 | 98.80±0.32 | 96.89±0.20 | 96.93±0.34 | 99.36±0.67 | 97.60±0.28 | 98.02±0.23 | 99.21±0.27 | 97.10±0.22 | 70.82±0.57 | 86.64±0.39 | 98.82±0.46 |
| COX2 | 52.15±0.16 | 52.48±0.38 | 53.34±0.27 | 49.56±0.11 | 49.11±0.26 | 48.61±0.60 | 59.68±0.41 | 59.38±0.69 | 57.81±0.50 | 58.93±0.47 | 59.81±0.30 | 72.35±0.58 | 64.13±0.23 | 64.36±0.16 | 63.19±0.31 | 54.73±0.15 | 51.86±0.81 | 59.99±0.20 |
| NCI1 | 51.39±0.19 | 50.22±0.12 | 54.18±0.67 | 50.41±0.31 | 45.11±0.33 | 61.88±0.23 | 43.33±0.25 | 46.44±0.60 | 69.46±0.36 | 65.29±0.21 | 75.75±0.47 | 74.32±0.34 | 68.32±0.22 | 69.13±0.58 | 61.58±0.40 | 58.74±0.18 | 49.94±0.78 | 45.96±0.65 |
| DHFR | 48.31±0.47 | 52.79±0.35 | 50.30±0.31 | 51.64±0.22 | 45.58±0.21 | 63.15±0.24 | 58.21±0.46 | 57.01±0.39 | 61.09±0.27 | 61.79±0.54 | 59.82±0.44 | 72.87±0.48 | 61.25±0.19 | 63.23±0.38 | 64.48±0.13 | 53.23±0.10 | 63.93±0.83 | 61.52±0.40 |
| IM→IB | 49.80±0.33 | 51.23±0.65 | 53.45±0.29 | 53.03±0.14 | 56.26±0.25 | 51.32±0.16 | 74.45±0.19 | 78.62±0.57 | 80.98±0.31 | 81.25±0.68 | 66.73±0.52 | 71.10±0.63 | 78.28±0.23 | 80.23±0.34 | 80.94±0.17 | 52.67±0.12 | 82.17±0.17 | 77.66±0.51 |
| EN→PR | 52.53±0.21 | 53.36±0.31 | 53.92±0.58 | 51.90±0.49 | 46.01±0.42 | 33.52±0.22 | 59.76±0.16 | 63.23±0.40 | 61.77±0.35 | 59.36±0.11 | 67.18±0.61 | 62.42±0.30 | 56.95±0.24 | 61.53±0.26 | 63.84±0.26 | 54.48±0.36 | 50.17±0.49 | 64.61±0.46 |
| AI→DH | 51.18±0.13 | 51.69±0.66 | 52.28±0.21 | 50.95±0.48 | 44.33±0.29 | 63.27±0.47 | 97.11±0.52 | 98.48±0.40 | 95.68±0.26 | 94.33±0.65 | 98.95±0.61 | 96.82±0.37 | 95.42±0.17 | 99.10±0.12 | 99.27±0.57 | 94.56±0.63 | 94.90±0.43 | 93.95±0.32 |
| BZ→CO | 43.34±0.58 | 52.43±0.14 | 49.76±0.47 | 52.16±0.54 | 64.29±0.20 | 64.65±0.31 | 78.98±0.44 | 76.01±0.62 | 87.27±0.32 | 80.55±0.69 | 81.86±0.29 | 89.11±0.37 | 83.21±0.66 | 96.32±0.17 | 95.16±0.49 | 65.26±0.24 | 77.44±0.63 | 80.97±0.26 |
| ES→MU | 52.99±0.25 | 52.63±0.13 | 52.13±0.20 | 52.28±0.42 | 58.12±0.26 | 51.57±0.65 | 78.66±0.17 | 79.66±0.32 | 86.70±0.36 | 90.55±0.47 | 88.32±0.44 | 91.43±0.69 | 89.30±0.55 | 92.41±0.14 | 91.98±0.39 | 75.65±0.12 | 91.91±0.90 | 83.92±0.21 |
| TO→SI | 53.73±0.34 | 51.87±0.67 | 53.50±0.18 | 52.25±0.64 | 64.32±0.22 | 66.53±0.60 | 66.85±0.38 | 64.85±0.57 | 67.29±0.29 | 69.80±0.50 | 68.91±0.52 | 66.72±0.33 | 72.51±0.26 | 68.24±0.47 | 66.70±0.13 | 72.34±0.15 | 66.90±0.63 | 67.21±0.65 |
| BB→BA | 54.15±0.61 | 53.11±0.48 | 54.62±0.15 | 53.48±0.12 | 63.27±0.39 | 32.37±0.30 | 69.18±0.22 | 67.33±0.37 | 78.83±0.46 | 77.69±0.14 | 78.93±0.60 | 89.88±0.25 | 79.07±0.17 | 80.17±0.44 | 81.44±0.11 | 55.69±0.67 | 72.00±0.63 | 75.94±0.13 |
| PT→MU | 51.52±0.12 | 55.87±0.19 | 54.03±0.26 | 52.12±0.40 | 55.88±0.13 | 53.78±0.15 | 78.10±0.58 | 79.74±0.65 | 79.27±0.18 | 77.54±0.66 | 62.51±0.50 | 84.63±0.24 | 80.12±0.27 | 79.44±0.36 | 82.05±0.47 | 58.28±0.61 | 67.65±0.44 | 80.42±0.24 |
| FS→TC | 50.06±0.55 | 54.76±0.25 | 51.98±0.53 | 53.24±0.64 | 44.98±0.27 | 49.57±0.66 | 67.05±0.22 | 66.01±0.30 | 66.98±0.13 | 68.92±0.36 | 64.38±0.18 | 78.12±0.60 | 67.32±0.29 | 69.89±0.41 | 71.58±0.50 | 60.12±0.38 | 67.65±0.41 | 67.09±0.26 |
| CL→LI | 50.85±0.47 | 51.74±0.55 | 52.66±0.30 | 51.54±0.18 | 55.62±0.45 | 56.45±0.11 | 59.65±0.33 | 54.17±0.48 | 61.21±0.27 | 58.31±0.50 | 59.30±0.37 | 72.15±0.69 | 63.42±0.21 | 70.21±0.14 | 69.28±0.29 | 50.79±0.66 | 53.08±0.72 | 60.93±0.54 |
| HIV-Size | 48.94±0.33 | 49.96±0.44 | 66.11±0.56 | 45.10±0.12 | 31.39±0.25 | 32.67±0.37 | 26.73±0.50 | 35.80±0.29 | 38.55±0.15 | 42.94±0.40 | 96.34±0.47 | 91.86±0.23 | 47.56±0.65 | 56.23±0.26 | 74.12±0.39 | 68.31±0.30 | 41.83±0.41 | 29.21±0.68 |
| ZINC-Size | 48.66±0.27 | 50.12±0.35 | 50.58±0.18 | 48.96±0.12 | 33.07±0.50 | 52.61±0.54 | 53.46±0.29 | 72.83±0.60 | 56.82±0.35 | 54.38±0.30 | 58.05±0.24 | 70.95±0.36 | 63.23±0.22 | 59.49±0.17 | 68.43±0.10 | 55.63±0.24 | 56.56±0.41 | 52.51±0.25 |
| HIV-Scaffold | 49.36±0.65 | 48.43±0.42 | 44.72±0.11 | 54.57±0.67 | 58.78±0.13 | 58.74±0.20 | 61.00±0.62 | 59.26±0.40 | 51.87±0.57 | 50.12±0.53 | 59.41±0.56 | 57.29±0.45 | 55.79±0.47 | 55.73±0.67 | 56.39±0.24 | 55.62±0.45 | 52.06±0.22 | 48.68±0.41 |
| ZINC-Scaffold | 51.12±0.50 | 46.66±0.26 | 51.17±0.34 | 53.28±0.65 | 54.77±0.14 | 55.17±0.38 | 54.04±0.22 | 55.80±0.21 | 42.58±0.41 | 41.29±0.52 | 97.36±0.13 | 81.43±0.45 | 39.63±0.30 | 50.37±0.56 | 50.71±0.29 | 45.24±0.47 | 40.57±0.51 | 27.78±0.12 |
| IC50-Size | 67.08±0.23 | 59.35±0.17 | 90.76±0.11 | 52.49±0.37 | 32.61±0.32 | 36.15±0.22 | 24.44±0.40 | 30.96±0.14 | 41.37±0.12 | 39.42±0.40 | 97.74±0.27 | 77.12±0.36 | 39.23±0.24 | 55.84±0.69 | 56.58±0.12 | 49.49±0.18 | 32.97±0.63 | |
| IC50-Scaffold | 66.33±0.49 | 60.95±0.55 | 88.49±0.12 | 58.56±0.30 | 36.96±0.28 | 38.67±0.66 | 34.60±0.21 | 38.52±0.33 | 44.56±0.22 | 42.97±0.57 | 96.00±0.60 | 77.58±0.47 | 38.43±0.10 | 57.96±0.19 | 56.62±0.51 | 59.68±0.40 | 49.79±0.42 | 36.34±0.63 |
| IC50-Assay | 64.95±0.22 | 62.78±0.38 | 86.03±0.18 | 55.27±0.63 | 27.88±0.34 | 30.36±0.60 | 31.28±0.23 | 32.35±0.41 | 51.92±0.19 | 48.43±0.50 | 94.18±0.14 | 74.65±0.44 | 40.98±0.30 | 66.52±0.11 | 58.29±0.56 | 54.24±0.29 | 45.85±0.62 | 41.86±0.57 |
| EC50-Assay | 49.08±0.41 | 46.66±0.65 | 48.43±0.23 | 45.32±0.19 | 47.80±0.44 | 47.81±0.66 | 47.80±0.55 | 46.02±0.52 | 56.11±0.21 | 52.44±0.28 | 69.31±0.31 | 82.97±0.30 | 54.34±0.13 | 66.71±0.40 | 65.57±0.17 | 58.96±0.11 | 60.42±0.40 | 55.79±0.52 |
| Avg. AUROC | 52.69 | 53.67 | 57.56 | 52.91 | 52.11 | 50.35 | 61.29 | 61.50 | 66.00 | 64.29 | 73.14 | 75.86 | 65.50 | 71.07 | 71.05 | 60.38 | 61.54 | 61.91 |
| Avg. Rank | 13.80 | 13.54 | 12.03 | 13.86 | 14.00 | 13.83 | 10.83 | 10.63 | 7.26 | 8.71 | 5.43 | 3.37 | 7.03 | 3.69 | 4.14 | 10.58 | 9.89 | 9.43 |

**AUROC (middle)**

| | GK-D (two-step) | | | | SSL-D (two-step) | | | | OCGIN | GLocalKD | OCGTL | SIGNET | GLADC | CVTGAD | GOOD-D | GraphDE | AAGOD | GOODAT |
|---|---|---|---|---|---|---|---|---|---|---|---|---|---|---|---|---|---|---|
| | PK-SVM | PK-iF | WL-SVM | WL-iF | IG-SVM | IG-iF | GCL-SVM | GCL-iF | | | | | | | | | | |
| p53 | 8.35±0.12 | 9.11±0.14 | 9.89±0.19 | 9.75±0.20 | 7.73±0.09 | 19.43±0.24 | 18.14±0.38 | 16.31±0.15 | 18.12±0.11 | 15.40±0.21 | 16.92±0.08 | 17.85±0.09 | 19.19±0.14 | 17.30±0.10 | 15.42±0.16 | 15.56±0.20 | 48.69±0.47 | 55.15±0.26 |
| HSE | 6.94±0.12 | 6.49±0.11 | 6.17±0.18 | 4.58±0.16 | 7.51±0.19 | 11.96±0.09 | 7.52±0.19 | 6.75±0.11 | 7.45±0.12 | 5.63±0.14 | 7.16±0.20 | 7.98±0.17 | 6.35±0.11 | 8.01±0.09 | 8.86±0.22 | 5.86±0.25 | 46.73±0.47 | 24.20±0.54 |
| MMP | 12.35±0.11 | 13.51±0.20 | 13.01±0.15 | 14.32±0.16 | 24.75±0.80 | 26.76±0.06 | 26.36±0.21 | 29.89±0.32 | 22.74±0.20 | 23.53±0.25 | 23.90±0.19 | 29.88±0.07 | 27.36±0.15 | 25.91±0.10 | 28.32±0.18 | 5.89±0.16 | 21.64±0.12 | 15.30±0.45 |
| PPAR | 6.44±0.12 | 5.16±0.13 | 4.79±0.19 | 5.58±0.10 | 14.88±0.27 | 5.75±0.15 | 10.83±0.11 | 11.83±0.25 | 9.30±0.14 | 9.25±0.20 | 9.01±0.22 | 11.39±0.18 | 11.57±0.13 | 9.09±0.12 | 10.70±0.14 | 8.63±0.17 | 49.98±0.11 | 18.72±0.17 |
| COLLAB | 43.76±0.25 | 41.98±0.20 | 43.12±0.19 | 52.15±0.25 | 46.05±0.09 | 43.53±0.10 | 46.29±0.21 | 56.97±0.32 | 47.89±0.25 | 46.11±0.18 | 69.24±0.40 | 58.47±0.22 | 70.59±0.35 | 63.90±0.29 | 46.28±0.17 | 59.92±0.61 | 60.07±0.67 | |
| IMDB-B | 41.58±0.23 | 40.42±0.21 | 49.04±0.19 | 59.29±0.22 | 48.51±0.22 | 45.71±0.25 | 63.78±0.30 | 58.02±0.24 | 60.34±0.18 | 54.49±0.20 | 58.45±0.21 | 67.12±0.34 | 61.72±0.26 | 68.89±0.35 | 66.15±0.30 | 57.23±0.27 | 57.94±0.19 | 68.30±0.42 |
| REDDIT-B | 45.22±0.21 | 47.55±0.24 | 61.20±0.30 | 74.00±0.34 | 20.59±0.12 | 6.47±0.10 | 82.60±0.40 | 79.40±0.29 | 95.63±0.36 | 81.47±0.28 | 89.16±0.33 | 85.46±0.31 | 79.81±0.25 | 88.23±0.27 | 91.72±0.34 | 49.05±0.16 | 54.68±0.75 | 74.60±0.60 |
| ENZYMES | 40.83±0.42 | 43.65±0.19 | 21.56±0.18 | 24.14±0.21 | 25.23±0.16 | 53.31±0.38 | 26.84±0.15 | 27.52±0.20 | 32.67±0.30 | 29.32±0.24 | 24.09±0.18 | 22.17±0.12 | 23.88±0.21 | 30.18±0.20 | 26.82±0.15 | 14.64±0.13 | 20.90±0.62 | 22.90±0.36 |
| PROTEINS | 44.07±0.20 | 31.38±0.19 | 43.39±0.23 | 48.02±0.27 | 70.12±0.32 | 18.69±0.14 | 78.00±0.35 | 77.88±0.34 | 79.78±0.37 | 76.98±0.30 | 70.85±0.28 | 75.21±0.20 | 81.93±0.40 | 83.92±0.35 | 80.19±0.30 | 62.13±0.22 | 79.54±0.12 | 77.80±0.28 |
| DD | 45.94±0.25 | 16.29±0.11 | 41.89±0.19 | 40.61±0.18 | 71.30±0.32 | 35.58±0.16 | 78.33±0.35 | 70.18±0.27 | 74.15±0.27 | 73.50±0.21 | 73.43±0.25 | 71.44±0.18 | 81.93±0.30 | 87.45±0.39 | 80.49±0.30 | 51.85±0.29 | 73.71±0.36 | 65.06±0.76 |
| BZR | 47.04±0.20 | 33.70±0.15 | 74.22±0.32 | 76.27±0.28 | 73.96±0.25 | 72.32±0.29 | 87.13±0.37 | 86.43±0.30 | 88.32±0.33 | 84.17±0.28 | 79.82±0.25 | 91.98±0.40 | 83.02±0.27 | 90.12±0.35 | 84.35±0.36 | 87.45±0.29 | 89.09±0.76 | 88.94±0.33 |
| AIDS | 42.46±0.23 | 41.30±0.22 | 10.97±0.11 | 29.69±0.15 | 60.48±0.27 | 92.44±0.40 | 82.12±0.37 | 96.47±0.38 | 93.42±0.37 | 95.28±0.35 | 97.89±0.40 | 63.58±0.20 | 89.43±0.31 | 96.35±0.38 | 85.05±0.28 | 15.23±0.14 | 72.72±0.25 | 49.30±0.19 |
| COX2 | 41.15±0.18 | 40.80±0.19 | 75.41±0.30 | 81.08±0.32 | 77.53±0.35 | 77.68±0.28 | 81.55±0.37 | 81.34±0.31 | 82.95±0.45 | 76.55±0.29 | 82.45±0.32 | 86.35±0.39 | 82.47±0.30 | 85.23±0.47 | 82.13±0.25 | 79.45±0.29 | 79.75±0.65 | 74.30±0.40 |
| NCI1 | 41.97±0.22 | 43.22±0.19 | 63.42±0.28 | 51.60±0.23 | 46.68±0.19 | 71.64±0.05 | 47.41±0.15 | 47.07±0.18 | 54.19±0.22 | 40.15±0.16 | 67.45±0.32 | 66.45±0.31 | 48.83±0.25 | 65.12±0.28 | 59.38±0.22 | 60.82±0.23 | 47.59±0.22 | 72.10±0.45 |
| DHFR | 45.28±0.20 | 40.48±0.18 | 30.43±0.15 | 34.42±0.19 | 28.85±0.16 | 34.87±0.20 | 44.35±0.22 | 42.49±0.19 | 47.32±0.25 | 38.92±0.21 | 38.69±0.29 | 58.64±0.35 | 44.65±0.23 | 32.43±0.30 | 59.88±0.25 | 47.31±0.27 | 59.30±0.18 | 57.00±0.16 |
| IM→IB | 43.68±0.20 | 42.14±0.17 | 37.60±0.15 | 49.26±0.19 | 54.58±0.22 | 48.14±0.13 | 67.99±0.27 | 69.63±0.21 | 76.34±0.24 | 76.84±0.25 | 62.50±0.19 | 67.98±0.16 | 71.59±0.22 | 74.35±0.21 | 71.12±0.19 | 38.45±0.16 | 78.58±0.53 | 74.40±0.40 |
| EN→MU | 40.74±0.18 | 39.85±0.17 | 50.45±0.19 | 63.07±0.22 | 52.90±0.20 | 44.97±0.16 | 65.68±0.19 | 67.23±0.22 | 67.45±0.28 | 66.98±0.21 | 73.95±0.28 | 67.50±0.19 | 61.25±0.20 | 70.78±0.24 | 68.72±0.22 | 50.90±0.18 | 59.03±0.46 | 65.00±0.71 |
| AI→DH | 42.19±0.20 | 41.64±0.18 | 35.28±0.15 | 31.47±0.13 | 46.20±0.20 | 67.59±0.27 | 95.72±0.30 | 98.02±0.01 | 92.55±0.27 | 89.25±0.25 | 97.85±0.30 | 93.45±0.25 | 92.01±0.24 | 96.30±0.28 | 99.12±0.35 | 89.67±0.26 | 93.47±0.33 | 92.50±0.21 |
| BZ→CO | 50.62±0.22 | 40.85±0.17 | 39.15±0.16 | 38.94±0.15 | 67.44±0.25 | 62.90±0.23 | 73.13±0.27 | 71.17±0.24 | 85.45±0.30 | 75.45±0.22 | 75.82±0.01 | 80.12±0.26 | 74.12±0.20 | 95.78±0.35 | 94.78±0.33 | 55.45±0.18 | 68.72±0.40 | 59.10±0.69 |
| ES→MU | 40.25±0.19 | 40.63±0.17 | 34.17±0.15 | 33.12±0.13 | 58.81±0.22 | 64.32±0.07 | 69.03±0.24 | 69.82±0.23 | 88.95±0.33 | 79.12±0.26 | 85.85±0.30 | 84.02±0.28 | 73.11±0.25 | 65.79±0.22 | 88.88±0.35 | 42.55±0.16 | 84.94±0.55 | 59.10±0.69 |
| TO→SI | 39.45±0.19 | 41.45±0.20 | 40.03±0.17 | 43.69±0.18 | 68.25±0.14 | 86.63±0.14 | 64.35±0.23 | 61.99±0.21 | 67.65±0.22 | 66.60±0.20 | 65.50±0.21 | 68.23±0.23 | 73.11±0.25 | 65.78±0.22 | 56.34±0.19 | 69.12±0.24 | 67.20±0.19 | 72.70±0.42 |
| BB→BA | 39.00±0.18 | 40.12±0.17 | 34.42±0.15 | 39.46±0.16 | 64.90±0.22 | 38.44±0.14 | 62.54±0.23 | 58.75±0.20 | 74.15±0.27 | 73.05±0.24 | 71.45±0.25 | 91.78±0.35 | 74.45±0.22 | 75.89±0.25 | 80.12±0.30 | 44.55±0.16 | 64.25±0.50 | 60.70±0.79 |
| PT→MU | 41.83±0.20 | 37.15±0.16 | 38.91±0.18 | 33.99±0.14 | 50.67±0.19 | 62.23±0.23 | 74.03±0.25 | 75.00±0.26 | 69.15±0.24 | 69.40±0.23 | 53.15±0.18 | 76.89±0.30 | 75.35±0.25 | 69.22±0.21 | 61.89±0.21 | 47.18±0.17 | 59.96±0.68 | 68.30±0.49 |
| FS→TC | 43.40±0.21 | 38.35±0.18 | 49.47±0.23 | 49.17±0.22 | 43.77±0.19 | 48.36±0.20 | 68.55±0.30 | 64.98±0.25 | 68.12±0.24 | 67.68±0.26 | 66.78±0.24 | 74.11±0.32 | 65.54±0.22 | 67.45±0.23 | 61.89±0.21 | 54.43±0.20 | 61.96±0.72 | 58.00±0.14 |
| CL→LI | 42.55±0.20 | 41.59±0.19 | 44.31±0.21 | 38.96±0.17 | 51.84±0.22 | 58.47±0.25 | 55.14±0.23 | 48.88±0.21 | 54.89±0.22 | 51.45±0.20 | 52.89±0.19 | 64.49±0.30 | 57.02±0.22 | 62.34±0.27 | 59.88±0.25 | 40.12±0.18 | 48.53±0.39 | 56.00±0.32 |
| HIV-Size | 50.61±0.25 | 43.27±0.21 | 61.71±0.28 | 51.63±0.24 | 38.67±0.19 | 39.42±0.18 | 37.08±0.17 | 40.25±0.20 | 36.67±0.16 | 31.73±0.14 | 88.78±0.35 | 94.12±0.40 | 36.96±0.19 | 52.23±0.22 | 71.43±0.30 | 38.45±0.16 | 38.85±0.58 | 36.60±0.16 |
| ZINC-Size | 51.02±0.22 | 45.78±0.20 | 51.02±0.21 | 50.11±0.19 | 52.18±0.20 | 52.86±0.23 | 51.50±0.19 | 52.10±0.20 | 49.95±0.40 | 50.83±0.20 | 56.23±0.24 | 54.14±0.22 | 49.60±0.30 | 51.50±0.21 | 69.04±0.45 | 50.07±0.19 | 45.10±0.46 | 43.33±0.74 |
| HIV-Scaffold | 50.19±0.22 | 47.83±0.20 | 50.39±0.19 | 55.48±0.23 | 56.71±0.24 | 56.61±0.27 | 57.73±0.25 | 56.88±0.23 | 54.67±0.22 | 54.20±0.20 | 56.24±0.23 | 66.57±0.30 | 56.83±0.22 | 55.31±0.20 | 57.22±0.21 | 49.89±0.19 | 49.97±0.70 | 52.33±0.43 |
| ZINC-Scaffold | 53.19±0.24 | 53.48±0.22 | 51.95±0.20 | 53.41±0.19 | 54.72±0.23 | 55.39±0.25 | 54.16±0.21 | 54.31±0.22 | 51.57±0.19 | 51.67±0.20 | 55.15±0.24 | 53.73±0.22 | 54.19±0.23 | 50.64±0.18 | 49.18±0.16 | 55.50±0.24 | 41.01±0.66 | 38.65±0.57 |
| IC50-Size | 65.92±0.30 | 61.15±0.28 | 87.46±0.35 | 59.71±0.25 | 38.88±0.19 | 40.42±0.20 | 36.47±0.17 | 38.67±0.18 | 55.81±0.40 | 42.92±0.20 | 88.95±0.40 | 75.26±0.30 | 35.13±0.16 | 51.88±0.23 | 50.76±0.24 | 60.89±0.50 | 57.56±0.24 | 54.42±0.13 |
| IC50-Scaffold | 66.34±0.28 | 63.74±0.27 | 88.35±0.34 | 64.61±0.25 | 41.79±0.19 | 42.32±0.20 | 41.57±0.19 | 42.69±0.21 | 49.81±0.23 | 44.67±0.21 | 93.74±0.40 | 73.14±0.31 | 41.77±0.18 | 57.36±0.24 | 56.38±0.23 | 52.33±0.20 | 50.75±0.18 | 61.40±0.39 |
| IC50-Assay | 67.09±0.29 | 67.08±0.28 | 87.74±0.34 | 63.28±0.25 | 36.86±0.18 | 38.06±0.19 | 39.37±0.20 | 38.46±0.19 | 49.81±0.23 | 44.67±0.21 | 93.74±0.40 | 73.14±0.31 | 41.77±0.18 | 57.36±0.24 | 51.80±0.19 | 50.02±0.17 | 53.05±0.48 | 51.23±0.14 |
| EC50-Assay | 53.75±0.23 | 53.15±0.22 | 57.65±0.25 | 53.38±0.20 | 50.45±0.19 | 50.19±0.18 | 51.74±0.19 | 50.95±0.20 | 50.67±0.21 | 48.47±0.18 | 64.82±0.30 | 57.66±0.26 | 50.84±0.19 | 57.27±0.23 | 85.26±0.32 | 50.38±0.46 | 51.23±0.14 | |
| Avg. AUPRC | 43.27 | 40.20 | 46.52 | 45.51 | 47.85 | 48.53 | 55.11 | 54.87 | 62.15 | 56.96 | 64.98 | 66.24 | 56.21 | 61.77 | 62.54 | 49.97 | 57.89 | 56.71 |
| Avg. Rank | 12.66 | 13.77 | 12.69 | 12.23 | 12.11 | 11.46 | 8.89 | 9.14 | 6.94 | 9.69 | 6.34 | 4.37 | 8.71 | 6.06 | 6.11 | 11.83 | 8.97 | 9.03 |

**FPR95 (bottom)**

| | GK-D (two-step) | | | | SSL-D (two-step) | | | | OCGIN | GLocalKD | OCGTL | SIGNET | GLADC | CVTGAD | GOOD-D | GraphDE | AAGOD | GOODAT |
|---|---|---|---|---|---|---|---|---|---|---|---|---|---|---|---|---|---|---|
| | PK-SVM | PK-iF | WL-SVM | WL-iF | IG-SVM | IG-iF | GCL-SVM | GCL-iF | | | | | | | | | | |
| p53 | 98.63±0.19 | 96.77±0.18 | 98.34±0.21 | 97.34±0.20 | 83.74±0.15 | 60.30±0.43 | 72.20±0.72 | 77.10±0.16 | 80.96±0.19 | 81.07±0.20 | 79.44±0.18 | 74.24±0.15 | 75.32±0.17 | 77.60±0.21 | 70.82±0.19 | 89.20±0.22 | 96.05±0.20 | 75.80±0.87 |
| HSE | 97.52±0.18 | 88.24±0.14 | 98.83±0.20 | 87.63±0.13 | 75.20±0.41 | 98.81±0.20 | 77.28±0.15 | 83.04±0.18 | 52.23±0.32 | 75.23±0.19 | 82.79±0.17 | 69.82±0.25 | 77.01±0.20 | 71.21±0.18 | 69.32±0.24 | 88.43±0.27 | 91.26±0.15 | 61.00±0.61 |
| MMP | 99.24±0.20 | 91.21±0.17 | 98.00±0.19 | 90.70±0.16 | 96.19±0.09 | 89.38±0.18 | 65.80±0.15 | 58.40±0.11 | 74.94±0.19 | 62.89±0.25 | 77.50±0.18 | 60.12±0.28 | 64.42±0.20 | 69.45±0.22 | 78.23±0.19 | 94.35±0.23 | 88.27±0.62 | 78.40±0.80 |
| PPAR | 98.15±0.19 | 97.33±0.18 | 98.45±0.20 | 95.95±0.19 | 85.23±0.16 | 75.88±0.55 | 82.70±0.18 | 84.29±0.11 | 86.87±0.22 | 87.90±0.23 | 77.06±0.20 | 86.41±0.21 | 91.88±0.16 | 86.11±0.17 | 85.02±0.30 | 96.33±0.19 | 83.50±0.50 | |
| COLLAB | 99.07±0.20 | 95.82±0.18 | 99.79±0.23 | 96.50±0.20 | 97.04±0.19 | 97.92±0.18 | 98.04±0.17 | 91.54±0.18 | 89.64±0.19 | 91.56±0.20 | 87.22±0.21 | 81.56±0.26 | 81.07±0.27 | 82.35±0.25 | 86.12±0.19 | 97.54±0.20 | 80.58±0.28 | 92.90±0.39 |
| IMDB-B | 98.61±0.19 | 88.32±0.16 | 99.00±0.22 | 89.80±0.17 | 96.40±0.21 | 94.07±0.20 | 79.20±0.16 | 84.00±0.19 | 87.89±0.20 | 95.03±0.22 | 83.21±0.18 | 72.50±0.25 | 78.00±0.24 | 75.34±0.27 | 82.90±0.17 | 94.62±0.21 | 90.79±0.30 | 97.90±0.20 |
| REDDIT-B | 99.36±0.23 | 81.97±0.16 | 98.00±0.20 | 82.90±0.17 | 84.00±0.14 | 94.60±0.21 | 53.40±0.12 | 49.20±0.11 | 33.55±0.35 | 36.59±0.30 | 50.38±0.19 | 52.11±0.21 | 50.92±0.18 | 45.87±0.17 | 44.77±0.09 | 96.36±0.24 | 54.68±0.75 | 98.72±0.38 |
| ENZYMES | 75.08±0.19 | 87.57±0.20 | 76.40±0.18 | 86.80±0.21 | 74.00±0.28 | 94.60±0.22 | 72.60±0.25 | 77.40±0.20 | 78.93±0.19 | 77.42±0.17 | 84.97±0.21 | 88.10±0.23 | 69.35±0.32 | 75.37±0.18 | 82.95±0.20 | 98.24±0.34 | 81.70±0.49 | 72.00±0.28 |
| PROTEINS | 69.16±0.23 | 76.26±0.21 | 79.78±0.25 | 96.89±0.22 | 60.67±0.20 | 80.00±0.17 | 78.44±0.17 | 79.11±0.19 | 67.82±0.32 | 69.52±0.19 | 92.22±0.21 | 75.36±0.20 | 68.20±0.17 | 72.41±0.20 | 68.10±0.28 | 90.45±0.23 | 83.50±0.19 | 96.00±0.31 |
| DD | 97.58±0.20 | 96.92±0.19 | 98.98±0.22 | 97.74±0.21 | 78.05±0.17 | 98.06±0.24 | 65.32±0.13 | 84.54±0.20 | 63.19±0.42 | 69.52±0.19 | 92.22±0.21 | 71.54±0.29 | 68.20±0.17 | 72.41±0.20 | 68.10±0.35 | 90.45±0.41 | 83.50±0.49 | 96.00±0.30 |
| BZR | 92.18±0.20 | 94.10±0.19 | 92.94±0.21 | 93.01±0.20 | 91.90±0.18 | 97.71±0.22 | 67.12±0.02 | 68.37±0.30 | 72.89±0.19 | 73.92±0.20 | 96.65±0.23 | 71.54±0.29 | 76.61±0.21 | 73.31±0.20 | 88.22±0.21 | 84.48±0.19 | 81.50±0.46 | 85.00±0.75 |
| AIDS | 98.40±0.20 | 58.22±0.12 | 99.78±0.22 | 58.75±0.14 | 38.12±0.10 | 5.12±0.03 | 15.13±0.07 | 2.25±0.02 | 13.57±0.08 | 14.30±0.09 | 1.40±0.01 | 24.98±0.10 | 6.79±0.05 | 4.22±0.03 | 12.65±0.07 | 93.09±0.21 | 97.06±0.17 | 84.42±0.12 |
| COX2 | 97.83±0.19 | 88.61±0.16 | 97.05±0.18 | 89.29±0.17 | 93.10±0.20 | 98.00±0.22 | 87.38±0.19 | 66.19±0.18 | 90.07±0.19 | 91.04±0.20 | 85.12±0.26 | 79.14±0.30 | 85.67±0.23 | 85.62±0.25 | 90.94±0.22 | 95.11±0.23 | 92.70±0.13 | 73.30±0.63 |
| NCI1 | 78.07±0.25 | 96.14±0.21 | 78.53±0.22 | 95.23±0.20 | 94.89±0.20 | 88.85±0.24 | 98.85±0.23 | 96.75±0.25 | 81.25±0.30 | 90.89±0.18 | 79.47±0.27 | 87.00±0.22 | 72.75±0.20 | 86.09±0.22 | 92.12±0.19 | 88.06±0.22 | 94.80±0.18 | 90.18±0.27 |
| DHFR | 97.14±0.19 | 93.30±0.18 | 98.20±0.21 | 93.70±0.19 | 95.23±0.20 | 75.00±0.30 | 79.36±0.26 | 84.36±0.23 | 85.21±0.20 | 78.84±0.14 | 94.03±0.21 | 83.38±0.19 | 82.76±0.20 | 81.73±0.25 | 84.50±0.23 | 88.34±0.19 | 89.50±0.22 | 92.13±0.05 |
| IM→IB | 99.08±0.24 | 92.37±0.06 | 99.78±0.05 | 93.33±0.03 | 77.14±0.07 | 94.51±0.09 | 67.73±0.08 | 46.00±0.24 | 47.01±0.07 | 36.67±0.29 | 77.21±0.07 | 77.64±0.05 | 49.84±0.06 | 53.52±0.03 | 46.61±0.26 | 97.73±0.06 | 41.43±0.34 | 42.50±0.43 |
| EN→PR | 97.71±0.05 | 88.73±0.32 | 98.00±0.06 | 90.00±0.28 | 98.67±0.04 | 98.86±0.09 | 92.67±0.03 | 91.00±0.05 | 94.18±0.07 | 98.92±0.06 | 90.30±0.25 | 94.81±0.07 | 99.26±0.09 | 90.58±0.03 | 93.67±0.08 | 97.93±0.06 | 98.39±0.84 | 89.40±0.50 |
| AI→DH | 92.83±0.05 | 98.47±0.04 | 93.30±0.03 | 99.40±0.06 | 81.95±0.03 | 88.00±0.02 | 10.10±0.02 | 6.90±0.01 | 11.92±0.02 | 11.65±0.01 | 4.45±0.01 | 6.18±0.02 | 7.86±0.02 | 3.85±0.01 | 6.42±0.02 | 44.65±0.07 | 15.18±0.37 | 53.60±0.65 |
| BZ→CO | 96.20±0.04 | 95.01±0.05 | 97.07±0.06 | 98.54±0.04 | 93.00±0.03 | 93.47±0.02 | 47.32±0.10 | 50.24±0.12 | 45.33±0.08 | 40.75±0.03 | 41.90±0.04 | 27.88±0.08 | 30.11±0.24 | 29.59±0.22 | 12.18±0.19 | 73.96±0.16 | 80.52±0.46 | 76.00±0.36 |
| ES→MU | 97.35±0.06 | 97.45±0.04 | 98.58±0.03 | 99.47±0.06 | 91.81±0.05 | 59.97±0.01 | 49.73±0.08 | 46.37±0.10 | 39.99±0.07 | 29.02±0.25 | 51.87±0.06 | 36.21±0.04 | 29.59±0.22 | 32.18±0.27 | 37.40±0.15 | 73.96±0.36 | 36.03±0.44 | 76.00±0.31 |
| TO→SI | 98.78±0.04 | 95.28±0.01 | 99.40±0.06 | 95.79±0.04 | 88.90±0.29 | 69.54±0.25 | 90.84±0.05 | 91.86±0.07 | 87.66±0.03 | 82.29±0.06 | 83.21±0.07 | 91.96±0.06 | 86.52±0.05 | 94.25±0.04 | 83.07±0.23 | 84.12±0.05 | 89.94±0.11 | 78.40±0.54 |
| BB→BA | 99.35±0.06 | 94.10±0.04 | 99.22±0.07 | 95.00±0.05 | 88.00±0.04 | 97.43±0.02 | 75.98±0.09 | 78.53±0.07 | 69.05±0.26 | 80.51±0.06 | 58.66±0.22 | 52.53±0.29 | 78.23±0.05 | 73.56±0.03 | 79.21±0.04 | 96.21±0.05 | 70.03±0.78 | 90.70±0.20 |
| PT→MU | 86.74±0.03 | 92.80±0.05 | 85.71±0.06 | 99.43±0.05 | 86.18±0.04 | 97.85±0.07 | 57.14±0.10 | 57.71±0.08 | 59.24±0.09 | 55.47±0.04 | 89.11±0.03 | 42.33±0.25 | 48.66±0.20 | 54.62±0.05 | 44.65±0.22 | 98.37±0.05 | 83.94±0.60 | 53.30±0.27 |
| FS→TC | 93.29±0.04 | 94.00±0.06 | 92.92±0.05 | 95.08±0.04 | 91.49±0.03 | 94.91±0.07 | 80.00±0.09 | 83.51±0.08 | 82.98±0.06 | 82.00±0.04 | 80.40±0.07 | 81.01±0.04 | 65.21±0.27 | 72.60±0.20 | 84.75±0.05 | 91.35±0.19 | 83.82±0.37 | 74.00±0.27 |
| CL→LI | 89.12±0.04 | 96.63±0.05 | 89.59±0.06 | 96.35±0.04 | 93.63±0.03 | 91.94±0.07 | 80.20±0.09 | 83.51±0.08 | 82.98±0.06 | 83.01±0.04 | 84.70±0.07 | 65.21±0.27 | 72.60±0.20 | 80.00±0.05 | 84.75±0.05 | 99.58±0.19 | 83.82±0.37 | 74.00±0.22 |
| HIV-Size | 94.92±0.05 | 93.48±0.06 | 75.80±0.03 | 98.68±0.04 | 90.04±0.05 | 98.72±0.04 | 99.40±0.06 | 96.52±0.05 | 99.68±0.07 | 99.92±0.06 | 4.56±0.02 | 59.40±0.28 | 99.16±0.08 | 98.80±0.06 | 84.36±0.04 | 93.24±0.05 | 97.23±0.71 | 99.80±0.14 |
| ZINC-Size | 99.04±0.06 | 93.76±0.05 | 96.00±0.04 | 95.88±0.03 | 93.04±0.05 | 91.76±0.29 | 92.32±0.26 | 92.44±0.25 | 94.88±0.07 | 95.68±0.06 | 92.48±0.05 | 93.12±0.07 | 93.12±0.06 | 97.12±0.08 | 99.08±0.07 | 95.68±0.06 | 86.60±0.77 | 98.88±0.22 |
| HIV-Scaffold | 98.92±0.05 | 98.56±0.06 | 98.80±0.05 | 94.00±0.04 | 91.16±0.03 | 99.24±0.04 | 86.64±0.28 | 85.80±0.25 | 89.88±0.05 | 94.04±0.04 | 91.32±0.03 | 79.96±0.30 | 87.72±0.07 | 95.32±0.06 | 92.80±0.05 | 91.60±0.04 | 96.21±0.66 | 98.20±0.17 |
| ZINC-Scaffold | 98.68±0.06 | 95.88±0.05 | 95.76±0.04 | 92.80±0.25 | 93.44±0.07 | 93.56±0.05 | 94.88±0.06 | 92.44±0.07 | 94.60±0.05 | 96.56±0.07 | 90.20±0.28 | 94.56±0.06 | 98.00±0.08 | 93.16±0.07 | 94.84±0.05 | 98.12±0.06 | 97.82±0.63 | 97.34±0.21 |
| IC50-Size | 83.84±0.04 | 96.04±0.06 | 29.84±0.26 | 96.72±0.06 | 94.04±0.05 | 99.20±0.06 | 98.84±0.07 | 98.76±0.05 | 98.72±0.07 | 99.24±0.06 | 11.44±0.02 | 69.68±0.25 | 95.88±0.08 | 96.64±0.07 | 97.52±0.06 | 96.96±0.05 | 98.48±0.44 | 99.00±0.68 |
| IC50-Scaffold | 75.96±0.06 | 95.76±0.07 | 27.36±0.25 | 98.44±0.06 | 99.20±0.08 | 98.92±0.05 | 99.96±0.06 | 98.56±0.05 | 98.72±0.07 | 99.20±0.06 | 4.51±0.01 | 70.76±0.25 | 96.60±0.08 | 97.52±0.07 | 97.52±0.06 | 96.96±0.04 | 99.80±0.89 | 90.00±0.68 |
| IC50-Assay | 85.28±0.05 | 94.88±0.04 | 48.48±0.25 | 94.76±0.05 | 98.28±0.06 | 98.20±0.05 | 98.68±0.07 | 97.48±0.06 | 98.76±0.07 | 99.28±0.08 | 15.36±0.02 | 70.76±0.28 | 97.80±0.26 | 85.80±0.06 | 97.16±0.05 | 95.72±0.07 | 89.18±0.52 | 99.20±0.68 |
| EC50-Assay | 93.48±0.05 | 94.72±0.06 | 91.04±0.25 | 94.36±0.05 | 96.48±0.07 | 95.76±0.04 | 96.00±0.06 | 95.36±0.05 | 96.08±0.07 | 97.80±0.26 | 82.80±0.05 | 89.24±0.08 | 84.88±0.07 | 89.68±0.06 | 91.60±0.05 | 91.52±0.10 | 90.08±0.63 | 98.12±0.41 |
| Avg. FPR95 | 94.02 | 92.76 | 88.88 | 93.84 | 89.39 | 89.02 | 77.13 | 76.73 | 75.66 | 76.60 | 66.40 | 69.62 | 74.59 | 73.72 | 75.02 | 91.23 | 83.45 | 81.59 |
| Avg. Rank | 13.17 | 12.03 | 13.09 | 12.69 | 11.77 | 11.63 | 8.43 | 7.46 | 8.17 | 9.66 | 6.57 | 4.54 | 7.29 | 7.09 | 7.20 | 12.03 | 9.54 | 8.66 |

