# OpenReview forum: "Unifying Unsupervised Graph-Level Anomaly Detection and Out-of-Distribution Detection: A Benchmark"
_ICLR.cc/2025/Conference — ICLR 2025 Poster_

### Official Review · Reviewer_cQQu · 2024-10-24

**Soundness:** 4
**Presentation:** 3
**Contribution:** 4
**Rating:** 8
**Confidence:** 4

**Summary:**

The paper introduces UB-GOLD (Unified Benchmark for unsupervised Graph-level OOD and anomaly Detection), a comprehensive benchmark that unifies two related tasks in graph machine learning: graph-level anomaly detection (GLAD) and graph-level out-of-distribution detection (GLOD). UB-GOLD bridges this gap by providing a unified evaluation across 18 representative methods and 35 datasets, covering various scenarios in anomaly and OOD detection. The benchmark offers a multi-dimensional analysis of methods, assessing their effectiveness, generalizability, robustness, and efficiency, while also providing an open-source codebase to facilitate reproducible research and encourage further exploration.

**Strengths:**

①The paper makes an original contribution by unifying GLAD and GLOD into a single benchmark, UB-GOLD. This creative combination highlights the conceptual overlap between the two tasks, simplifying evaluation in both areas.

②The paper's experiments evaluate 18 methods on 35 datasets. It offers a multi-dimensional analysis. The breadth of datasets used ensures reliable, real-world applicability, while the open-source codebase.

③UB-GOLD offers insights into detecting near-OOD samples and noisy training data. These findings current limitations in existing methods. What's more, this work focus on generalizability, robustness, and efficiency ensures the benchmark’s practical relevance across different application domains, making it a useful resource in the field.

**Weaknesses:**

①While the paper provides extensive comparisons across methods, it could benefit from a more in-depth discussion of why certain methods fail in specific scenarios. For instance, understanding the root causes of poor performance under noisy training data or near-OOD conditions.

②Consider adding a more concise summary section that highlights the most important findings from your multi-dimensional analyses. This could be in the form of a bulleted list or a short paragraph at the end of each results subsection to help readers quickly digest key insights.

③In datasets like IC50-Size, the out-of-distribution samples differ primarily in graph structure from the in-distribution ones. It’s unexpected that SSL-D and GK-D methods perform poorly on these datasets. Could the authors offer an explanation for these results?

**Questions:**

①Performance of SSL-D and GK-D on Graph Structure-Based Datasets: In datasets like IC50-Size, where out-of-distribution samples primarily differ from in-distribution ones in graph structure, SSL-D and GK-D methods show poor performance. Could you provide insights into why these methods struggle with such datasets? Is it related to the specific way these methods handle graph structure?

②Handling Near-OOD Samples: You mention that near-OOD samples are more difficult to detect than far-OOD samples. Could you elaborate on what specific characteristics of near-OOD samples make them harder to distinguish? Do you have any thoughts on how future work could address this issue?

**Details Of Ethics Concerns:**

See above.

---

> ### Author Response · Authors · 2024-11-21
> **Response Reviewer cQQu (1/2)**
>
> **W1: While the paper provides extensive comparisons across methods, it could benefit from a more in-depth discussion of why certain methods fail in specific scenarios. For instance, understanding the root causes of poor performance under noisy training data or near-OOD conditions.**
>
> **Answer W1:** Thank you for your valuable suggestion. While the primary goal of our benchmark is to provide a robust framework and dataset setting for GLAD/GLOD research, we agree that a deeper discussion of method performance in specific scenarios, such as near-OOD detection and noisy training data, would enhance the paper. Below is a more detailed analysis based on our observations:
>
> * **Near-OOD Samples Are More Difficult to Detect than Far-OOD Samples.** Figure 4 demonstrates that models generally struggle more with near-OOD samples compared to far-OOD ones. End-to-end methods like GLocalKD, OCGTL, and SIGNET consistently achieve higher AUROC values for far-OOD scenarios, especially on datasets such as AIDS, BZR, and ENZYMES. The key challenge with near-OOD samples lies in their high similarity to ID data, which requires models to capture subtle deviations that are often overlooked. This highlights the importance of developing methods capable of fine-grained feature discrimination.
>
> * **Limited Generalizability in Specialized Scenarios.** Significant performance gaps emerge between near-OOD and far-OOD conditions, particularly in specialized scenarios. For instance, methods such as GOOD-D and GraphDE show clear limitations when dealing with near-OOD samples, where their detection capabilities are compromised due to the overlap with ID-like features. Similarly, GLAD methods often falter in size-based anomaly detection tasks, indicating that their design may not fully account for certain domain-specific characteristics or structural shifts.
>
> * **Dataset-Specific Sensitivity Variations.** Our findings also reveal that the sensitivity of methods to anomalies varies significantly across datasets. For example, models like GLocalKD and OCGTL excel on the AIDS dataset but perform inconsistently on others, such as ENZYMES, where dataset-specific complexities reduce their effectiveness. This variability underscores the need for benchmarks like ours to evaluate methods across a diverse set of datasets, exposing strengths and weaknesses in different contexts.
>
> These observations emphasize the critical need for more robust and adaptable models that can effectively handle near-OOD detection challenges, noisy training data, and diverse dataset characteristics. While the main manuscript focused on presenting the benchmark framework, these nuanced findings underline its value in highlighting gaps in current methods and guiding future research. We hope this deeper analysis provides the clarity and context you suggested. Thank you again for your insightful feedback.
>
>
> **W2: While the paper provides extensive comparisons across methods, it could benefit from a more in-depth discussion of why certain methods fail in specific scenarios. For instance, understanding the root causes of poor performance under noisy training data or near-OOD conditions.**
>
> **Answer W2:** Thank you for your valuable suggestion. We recognize the importance of providing concise summaries to enhance the readability and accessibility of our findings. In response, we have added summary sections at the end of each results subsection to highlight the most important insights from our multi-dimensional analyses. These summaries are presented in a bulleted list format to allow readers to quickly grasp the key takeaways.

---

> > ### Author Response · Authors · 2024-11-21
> > **Response Reviewer cQQu (2/2)**
> >
> > **W3 and Q1:Performance of SSL-D and GK-D on Graph Structure-Based Datasets: In datasets like IC50-Size, where out-of-distribution samples primarily differ from in-distribution ones in graph structure, SSL-D and GK-D methods show poor performance. Could you provide insights into why these methods struggle with such datasets? **
> >
> > **Answer W3 and Q1:** Thank you for this insightful question regarding the performance of SSL-D and GK-D methods on graph structure-based datasets, particularly in Type IV datasets like IC50-Size, where OOD samples differ primarily in graph structure. Below is an enhanced analysis of why these methods struggle, focusing on their inherent limitations and design choices:
> >
> > * **SSL-D Method:** SSL-D methods rely on self-supervised learning techniques, such as InfoGraph (IG) or GraphCL (GCL), to generate graph embeddings, which are then used for downstream classification. While these embeddings are effective for datasets in Types I, II, and III, where ID and OOD samples differ clearly in label-related attributes, they are less effective for Type IV datasets, which involve structural distinctions (e.g., Size, Scaffold, or Assay).  In Size-based tasks, OOD samples are distinguished primarily by differences in graph size. SSL-D methods focus on learning general graph representations without explicitly capturing size-specific nuances. This lack of task-awareness hinders their ability to distinguish between ID and OOD samples effectively. The embeddings learned by SSL-D are task-agnostic, which works well in scenarios with clear label-based shifts but fails in cases requiring fine-grained understanding of structural properties.
> >
> > * **GK-D Method:** GK-D methods take a fundamentally different approach, focusing primarily on graph structural features while largely ignoring task-specific attributes. This enables GK-D to better capture structural differences, making it relatively more effective on Type IV datasets compared to SSL-D.  While GK-D is better suited for tasks involving Size, Scaffold, or Assay distinctions, its simplicity and reliance on traditional classifiers limit its performance. It lacks the adaptability of end-to-end (E2E) models, which can jointly optimize representation learning and downstream classification. GK-D methods often over-rely on handcrafted features and predefined metrics, which are insufficient for complex, nuanced structural variations in OOD samples.
> >
> > SSL-D methods struggle on graph structure-based datasets due to their inability to capture subtle structural distinctions, particularly size-specific variations. GK-D methods, while more effective at handling structural tasks, are constrained by their reliance on traditional classifiers and lack of integrated optimization. These findings highlight the need for advanced models that can combine robust structural representation learning with task-specific adaptability, enabling better performance in nuanced OOD scenarios like those in Type IV datasets.
> >
> >
> > **Q2: Handling Near-OOD Samples: You mention that near-OOD samples are more difficult to detect than far-OOD samples. Could you elaborate on what specific characteristics of near-OOD samples make them harder to distinguish? Do you have any thoughts on how future work could address this issue?**
> >
> > **Answer Q2:** Thank you for this insightful question. The challenge with detecting near-OOD samples lies in their high similarity to in-distribution (ID) data, which makes them much harder to distinguish compared to far-OOD samples. Unlike far-OOD samples that exhibit clear deviations in structure or attributes, near-OOD samples often share overlapping features with ID data and may fall close to or within the decision boundary, leading to increased uncertainty. For example, in molecular datasets, near-OOD samples might have similar chemical scaffolds or functional groups to ID samples, despite belonging to different distributions. This subtlety complicates the task of drawing precise boundaries between ID and near-OOD data.
> >
> > To address this challenge, future work could focus on developing models that enhance feature discrimination and robustness. Methods that employ fine-grained feature representation learning or hierarchical embeddings can help capture subtle deviations between ID and near-OOD samples. Additionally, techniques like adaptive decision boundaries and contrastive training with hard negatives may improve the ability to distinguish between these closely related samples. Incorporating uncertainty modeling, such as Bayesian approaches or ensemble methods, can also aid in better calibrating confidence scores for near-OOD detection.
> >
> > In summary, the inherent subtlety of near-OOD samples demands models that can precisely capture nuanced differences while maintaining robustness in decision-making. Tackling this challenge will be crucial for advancing OOD detection methods. Thank you again for highlighting this important aspect.

---

> > > ### Comment · Reviewer_cQQu · 2024-11-25
> > >
> > > Dear authors, I appreciate the thorough clarification provided, which has satisfactorily addressed the majority of my concerns. I will uphold my positive evaluation.

---

> > > > ### Author Response · Authors · 2024-11-25
> > > >
> > > > Dear Reviewer cQQu,
> > > >
> > > > Thank you for your thoughtful feedback and for taking the time to review our clarifications. We sincerely appreciate your positive evaluation and are glad that our responses have satisfactorily addressed your concerns. Your insights have been invaluable in helping us refine our work.
> > > >
> > > > Best regards,
> > > >
> > > > The Authors

---

### Official Review · Reviewer_yW1o · 2024-10-29

**Soundness:** 3
**Presentation:** 3
**Contribution:** 3
**Rating:** 6
**Confidence:** 4

**Summary:**

This paper unifies the unsupervised  graph level ood and anomaly detection as generalized graph level ood detection problem and conduct a comprehensive and unified benchmark. This paper provide some remarkable observation through the comprehensive comparison and analyses.

**Strengths:**

1. A large amount of datasets are employed to establish the benchmark to fairly compare the mutiple GLAD and GLOD emthods under a unified experimental setting.
2. In addition to the performance, the paper also investigates more characteristics of methods including generalizability, robustness, and efficiency.
3. The comprehensive benchmark and the multiple observations are provided in the experimental analysis to inspire future work.
4. There is no clear distinction between the training set and test set in unsupervised GLAD. Leveraging unsupervised graph-level anomaly detection methods for OOD detection is a good point.

**Weaknesses:**

1. In GLAD, anomalies can be considered as out-of-distribution (OOD) test samples for type 1 and type 2, but further clarification is needed in the paper on how the in-distribution (ID) test samples are constructed.
2. In an unsupervised setting, does this mean the test set can also be used as the training set? The in-distribution samples in the test set could potentially help improve performance.
3. From the several observations, we found there are large overlaps between OOD samples,  in addition to the experimental explanation, more specific analysis is needed on this point for particular datasets.

**Questions:**

1. What is the difference between near-OOD samples and anomalies if both come from the same dataset?
2. If some OOD samples or anomalies are integrated into the training set as contamination, are they still retained in the test set during inference?

---

> ### Author Response · Authors · 2024-11-21
> **Response Reviewer yW1o (1/2)**
>
> We appreciate Reviewer  yW1o for the positive review and constructive comments. We provide our responses as follows.
>
> **W1: In GLAD, anomalies can be considered as out-of-distribution (OOD) test samples for type 1 and type 2, but further clarification is needed in the paper on how the in-distribution (ID) test samples are constructed.**
>
>
> **Answer W1:** This is an excellent question, and we appreciate your attention to this detail. We have supplemented the description of in-distribution (ID) test samples in Appendix B: Detailed Description of Datasets in UB-GOLD. Below is a summary of how ID samples are constructed for the two dataset types:
> * Type I: Datasets with Intrinsic Anomaly. These datasets inherently include both ID and OOD samples. For the test set, ID samples are selected proportionally from different labels, ensuring diversity in the in-distribution data. For four of these datasets, the ratio of test_{ID} to train_{ID} varies significantly. This design choice allows models to be tested under various conditions, ensuring their robustness in distinguishing ID and OOD samples.
> * Type II: Datasets with Class-Based Anomaly. In this case, the ratio of test_{ID} to train_{ID} is fixed at 1:4. We employ 5-fold cross-validation during the experimental process, ensuring that all data points are used for both training and testing in a balanced manner.
>
> **W2：In an unsupervised setting, does this mean the test set can also be used as the training set? The in-distribution samples in the test set could potentially help improve performance.**
>
> **Answer W2:** Thank you for raising this important question. In our benchmark, the test set samples are strictly separated from the training set and are not used during training to ensure unbiased evaluation. Allowing test set samples to be included in the training data could lead to data leakage, artificially inflating performance and compromising the integrity of the benchmark.
>
> In our setup, the training set consists solely of in-distribution (ID) samples, enabling the model to learn the characteristics of the ID data without being influenced by the test set. The test set, on the other hand, includes both ID and OOD samples, providing a robust evaluation of the model’s detection capabilities. This separation reflects real-world scenarios where unseen data must be detected independently and adheres to best practices in OOD detection and anomaly detection tasks. We believe this approach ensures that the reported performance metrics are reliable and reflective of the model's true generalization ability.
>
>
> **W3:From the several observations, we found there are large overlaps between OOD samples, in addition to the experimental explanation, more specific analysis is needed on this point for particular datasets.**
>
> **Answer W3:** Thank you for your question. We strictly ensured that the dataset partitions are non-overlapping, meaning there is no overlap between OOD samples and ID samples, or between training and test sets. For each dataset, we carefully curated the splits to maintain a clear separation, ensuring that OOD samples are entirely distinct from ID data and that this separation is consistently upheld across all experimental setups.
>
> We sincerely apologize for any misunderstanding caused by our phrasing in the manuscript. To address this, we will add a specific statement in the main text(page 15) explicitly clarifying that there is no overlap in the dataset partitions. This clarification ensures that readers are fully aware of our rigorous dataset preparation process.
>
> By maintaining strict non-overlapping partitions, we eliminate any potential data leakage and provide results that reliably reflect the model’s ability to generalize to truly unseen OOD samples. We appreciate your observation and hope this explanation resolves your concern.

---

> > ### Author Response · Authors · 2024-11-21
> > **Response Reviewer yW1o (2/2)**
> >
> > **Q1: What is the difference between near-OOD samples and anomalies if both come from the same dataset?**
> >
> > **Answer Q1:** Thank you for this valuable question. The key difference lies in the task design and how the datasets are partitioned. In the near-far OOD detection task, as opposed to the main OOD/AD detection task, we specifically re-split the datasets to create distinct groups: test_{ID},test_{near-OOD} and test_{far-OOD}. Each of these subsets is carefully curated to have an equal number of samples, ensuring a balanced evaluation. This setup allows us to better analyze the model’s ability to distinguish between different types of samples and to determine whether its performance varies significantly for near-OOD, far-OOD, or ID samples. For more details on the dataset splits and methodology, please refer to Table 5 in the appendix. We hope this explanation clarifies the distinction and provides further insight into the experimental design.
> >
> > **Q2: If some OOD samples or anomalies are integrated into the training set as contamination, are they still retained in the test set during inference?**
> >
> > **Answer Q2:** No, they are not retained in the test set during inference. We accounted for this scenario by pre-removing OOD samples equivalent to 30% of ID_{train} from the OOD samples before integrating them into ID_{train} as contamination. These OOD samples were then gradually added to the training set in batches to simulate contamination scenarios. The detailed partitioning and process are described in Table 6 of the appendix. This ensures that there is no overlap between the contaminated training set and the test set, maintaining the integrity of the evaluation.

---

> > > ### Comment · Reviewer_yW1o · 2024-11-24
> > >
> > > Dear authors
> > > I truly appreciate the detailed clarification, which has addressed most of my concerns. I will maintain my positive score.

---

> > > > ### Author Response · Authors · 2024-11-24
> > > >
> > > > Dear Reviewer yW1o,
> > > >
> > > > Thank you for your kind and constructive feedback. I truly appreciate your positive evaluation and the time you have taken to review our work. Your insights are invaluable, and we will continue to refine our manuscript based on the feedback provided.
> > > >
> > > > Best regards,
> > > >
> > > > The Authors

---

### Official Review · Reviewer_2xER · 2024-11-03

**Soundness:** 4
**Presentation:** 4
**Contribution:** 3
**Rating:** 6
**Confidence:** 4

**Summary:**

This manuscript presents a unified benchmark called UB-GOLD, which aims to unify and benchmark Graph Level Anomaly Detection (GLAD) and Graph lavel OOD detection (GLOD). Specifically, the authors compare the performance of 18 GLAD/GLOD methods on 35 datasets. The performance is investigated in four different dimensions: effectiveness (using three common accuracy metrics), generalisability, robustness, and efficiency (time and memory usage). Based on extensive experiments, the authors provide insightful observations and discuss possible future directions.

**Strengths:**

1. this manuscript is well written and easy to follow;
2. I believe this is the first work that attempts to unify and benchmark GLAD and GLOD tasks, the findings could be broadly interesting to the graph data mining community;
3. the authors conducted extensive experiments and open-sourced their code base (which I believe is easy to extend), making contributions to future research;

**Weaknesses:**

## Major Comments

### 1. Unification and Benchmarking of GLAD and GLOD Tasks
As the first work attempting to unify (please indicate the related work if this is not true) and benchmark GLAD (Graph-Level Anomaly Detection) and GLOD (Graph-Level Out-of-Distribution) detection tasks, I believe the authors should dedicate more space and effort to this unification process. Specifically:

- **1.1) Formal Definitions**: It would be beneficial for the authors to formally (with math symbols) define GLAD and GLOD before introducing the concept of "Unsupervised generalized graph-level OOD detection." Providing these definitions upfront would establish clarity and context for readers unfamiliar with these tasks.

- **1.2) Relationship Between GLAD and GLOD**: The authors should discuss the relationship between GLAD and GLOD in more depth, as there seems to be room for clarification on whether they perceive OOD detection as a broader concept than anomaly detection. In my view, anomaly detection is a broader concept, as abnormal instances may either (1) exist within the distribution but in low-density regions or (2) represent out-of-distribution instances. Including a specific section comparing and contrasting GLAD and GLOD, with formal definitions and discussions of their conceptual overlap and distinctions, would enhance the unification effort and clarify this relationship.

---

### 2. Experimental Details and Completeness
There are some missing details regarding the experiments and results. Adding these would make the manuscript more self-contained, especially as it aims to serve as a benchmark paper.

- **2.1) Definition and Generation of Anomalies/OOD Samples**: To make the manuscript self-contained, please provide details in the appendix on how the anomalies or OOD samples are defined or generated in each dataset. Clear explanations will help readers replicate and understand your experiments.

- **2.2) Completeness of Results in Figures**: In Figure 4, results are only provided for 15 out of 19 methods, with similar omissions in Figures 5 and 6. If these omissions are due to space limitations, including the full results in the appendix would help ensure completeness. Please also add a brief discussion explaining why certain methods were omitted from the main figures, if applicable.

- **2.3) Metrics and Additional Results**: In Figures 4 and 5, please specify which metrics are being used. Additionally, including complete results for all metrics in the appendix, along with brief analyses, would make the benchmark more comprehensive and allow for a better understanding of the methods' performances.

---

### 3. Clarification of Experimental Settings and Terminology

- **3.1) Hyperparameter Search**: In the hyperparameter search section, the authors conduct a random search to optimize hyperparameters based on **test set performance**. This approach raises questions regarding practical applicability. Could you clarify if a validation set was used instead of the test set for hyperparameter tuning, or if there was a specific reason for using the test set? Also, discussing the implications of this choice on the generalizability of your results would provide valuable context.

- **3.2) Use of the Term "Generalisability"**: The term "generalisability" is typically associated with a model’s performance on unseen data, while the manuscript uses it to describe differences in OOD detection for near- and far-OOD samples. Alternative terms, such as "OOD detection range" or "OOD sensitivity spectrum," may better capture the intended meaning. Clarifying or refining this terminology would enhance reader understanding.

---

## Minor Comments and Suggestions

- **4.1) Page 4, Line 084: Typo**: Please correct "28 GLAD and GLOD methods" to "18 GLAD and GLOD methods."

- **4.2) Anomaly Ratio in Table 1**: In Table 1, adding the anomaly ratio for each dataset would make the data overview more informative.

- **4.3) Visualization of CPU and GPU Results in Figure 6**: In Figure 6, consider superimposing the CPU and GPU bars (distinguished by color) to make the visual comparison clearer.

- **4.4) OOD Judge Score Distribution Analysis**: On Page 5, you mention an "OOD Judge Score Distribution Analysis," but this concept was not referenced later in the main text, though it appears in the appendix. Including a reference in the main text would clarify its relevance.

- **4.5) "Well-trained GNNs" on Page 5**: On Page 5, the term "well-trained GNNs" is used. If this refers to pre-trained GNNs, please provide a brief explanation for clarity.

---

## Conclusion
Overall, I appreciate the authors’ efforts in developing a unified framework and benchmark for GLAD and GLOD tasks. Addressing these points would improve the manuscript’s clarity, depth, and completeness, making it a more robust resource for the community. And I will consider increasing my rating if they (or some of them) are well addressed.

---

**Questions:**

Please check the weak points.

---

> ### Author Response · Authors · 2024-11-21
> **Response Reviewer  2xER (1/2)**
>
> We are grateful to Reviewer 2xER for providing insightful feedback. The detailed responses are provided below.
>
> **W1: Unification and Benchmarking of GLAD and GLOD Tasks.**
> We sincerely appreciate the reviewer’s detailed feedback on the unification and benchmarking of GLAD and GLOD tasks. Below, we address each point raised and provide clarifications and proposed enhancements to improve the manuscript:
>
> **W1.1: Formal Definitions, providing these definitions upfront would establish clarity and context for readers unfamiliar with these tasks.**
>
> **Answer W1.1:** We acknowledge the importance of providing formal definitions for GLAD and GLOD before introducing the unified framework of "Unsupervised generalized graph-level OOD detection." These definitions will help readers unfamiliar with these tasks understand their foundations and provide necessary context for the unification effort.
>
> In the revised manuscript (pages 3 and 19), we have included detailed formal definitions of GLAD and GLOD using mathematical symbols. This step ensures that the foundational concepts of GLAD and GLOD are clearly established before introducing the unified task of unsupervised generalized graph-level OOD detection, providing readers with a structured and logical progression of ideas. These updated sections are highlighted in blue for easy reference.
>
>
> **W1.2: Relationship Between GLAD and GLOD.**
>
> **Answer W1.2:** To clarify the relationship between GLAD and GLOD, we focus on their objectives and shared underlying principles rather than framing one as broader than the other. While the two tasks may have conceptual differences, their goals are aligned in practice: identifying graphs that deviate from the expected norm.
>
> Both tasks ultimately aim to separate graphs into two categories: graphs that align with the expected norm and graphs that do not. This shared objective forms the foundation for our unified task of **generalized graph-level OOD detection**, which treats both anomalies and OOD graphs as generalized OOD samples. By unifying these tasks, we simplify their conceptual overlap, enabling methods and evaluations to address both scenarios under a single framework.
>
> **W2: Experimental Details and Completeness.**
>
> **W2.1: Definition and Generation of Anomalies/OOD Samples.**
>
> **Answer W2.1:** We have added detailed explanations regarding the definition and generation of anomalies and OOD samples in **Appendix B: Detailed Description of Datasets in UB-GOLD**. This section includes comprehensive descriptions for each type dataset used in the benchmark.
>
> **W2.2: Completeness of Results in Figures.**
>
> **Answer W2.2:** We sincerely thank you for raising such a detailed and thoughtful question. Below, we provide explanations for the omissions in Figures 4 and 5 and outline the steps we’ve taken to address this concern:
>
> * Omission of GK-D Method: The GK-D method differs from other approaches as it requires incorporating the test set into the training process to achieve meaningful results. This design makes it incompatible with tasks such as near-far OOD detection and perturbation analysis, which involve finer-grained partitioning of the test or training set. As this method's behavior does not align with the experimental setups for these tasks, we did not include GK-D in Figures 4 and 5.
>
> * Omission of SSL-D Method in Figure 5: The SSL-D method exhibited relatively poor performance in the experiments. Including its results in Figure 5 would make them visually indistinguishable and potentially reduce the clarity of the figure.
>
> * Complete Results for Figure 5 Added to the Appendix:
> We have supplemented the full results for Figure 5 in the appendix (Figures 8–10). These results ensure that readers have access to all experimental outcomes for transparency and completeness.
>
>
> **W2.3:Metrics and Additional Results:**
>
> **Answer W2.3:** We sincerely thank you for highlighting this important aspect. In the main text, we used the AUROC metric to analyze the results presented in Figures 4 and 5, as it is a widely recognized standard for evaluating model performance. To further clarify this point, we have updated the captions for Figures 4 and 5 to explicitly state that the results are presented in terms of AUROC. In the original manuscript, we included results for two additional metrics, AUPRC and FPR95, for Figure 4 in Table 7 of the appendix. To address the reviewer’s suggestion, we have supplemented the revised manuscript with complete results for all metrics (AUROC, AUPRC, and FPR95) related to Figure 5 and included them in the appendix as Figures 8–10. Additionally, we have added brief analyses in the appendix to provide a detailed comparison of methods across all metrics, offering readers a more comprehensive understanding of the methods’ performances. We hope these updates enhance the clarity, completeness, and value of the manuscript as a benchmark resource.

---

> > ### Author Response · Authors · 2024-11-21
> > **Response Reviewer 2xER (2/2)**
> >
> > **W3: Clarification of Experimental Settings and Terminology.**
> >
> > **W3.1 Hyperparameter Search.**
> >
> > **Answer W3.1:** We sincerely thank you for raising this critical and insightful question. The decision to optimize hyperparameters based on test set performance, rather than using a validation set, was made deliberately after extensive discussions among the authors. The reasons are as follows.
> >
> > In tasks like OOD detection and anomaly detection, using a validation set can be problematic due to the potential distributional mismatch between the validation and test sets. Specifically, there is no guarantee that a model performing well on a validation set will generalize effectively to the test set, as their distributions may differ significantly. This issue has also been recognized in related fields such as **computer vision (CV)**. For example, benchmarks like **OpenOOD[1,2]** often rely on test set performance for hyperparameter tuning, reflecting the challenges of ensuring representative validation sets in these tasks.
> >
> > In real-world scenarios, this misalignment poses a significant risk, as models optimized for a mismatched validation set could perform poorly on unseen test distributions. Such a situation would lead to catastrophic outcomes in practical applications, where the goal is to handle diverse and unknown OOD distributions.
> >
> > Instead, we chose to focus on training ID models that exhibit robust generalization capabilities across a wide range of OOD scenarios. By directly evaluating hyperparameters on the test set, we aim to identify models that can perform well in real-world tasks where test distributions are often unknown and validation data might not be representative. This approach reflects the practical reality of OOD detection tasks and aligns with our goal of creating a benchmark that emphasizes robust, real-world performance.
> >
> > We hope this explanation clarifies our design choice and its alignment with both practical applicability and established practices in the field.
> >
> > [1] Yang J, Wang P, Zou D, et al. Openood: Benchmarking generalized out-of-distribution detection[J]. Advances in Neural Information Processing Systems, 2022, 35: 32598-32611.
> >
> > [2] Zhang J, Yang J, Wang P, et al. Openood v1. 5: Enhanced benchmark for out-of-distribution detection[J]. arXiv preprint arXiv:2306.09301, 2023.
> >
> > **W3.2  Use of the Term "Generalisability".**
> >
> > **Answer W3.2:** We sincerely thank you for your constructive and thoughtful feedback. After careful discussion, we agree with your observation that the term "generalizability" is not entirely appropriate when describing differences in OOD detection performance for near- and far-OOD samples. We also greatly appreciate your suggestion of the term **"OOD sensitivity spectrum"** which better captures the intended meaning. Following your advice, we have made comprehensive revisions to the manuscript, replacing "generalizability" with "OOD sensitivity spectrum" where applicable. We believe this change enhances clarity and improves reader understanding. Thank you again for your valuable input.
> >
> >
> >
> > **W4: Minor Comments and Suggestions.**
> >
> > **Answer W4:** We have addressed each point as follows and made the corresponding revisions in the manuscript, highlighted in blue for clarity:
> > 1. The typo "28 GLAD and GLOD methods" has been corrected to "18 GLAD and GLOD methods."
> > 2. We have added the anomaly ratio for each dataset in Table 1 to provide a more informative overview of the datasets.
> > 3. We sincerely thank the reviewer for the thoughtful suggestion regarding the visualization in Figure 6. Based on your feedback, we experimented with superimposing the CPU and GPU bars using distinct colors. However, we found that this approach significantly reduced the clarity of the figure, especially given the visual complexity of comparing multiple methods. Additionally, since the GK-D method does not utilize GPUs, its inclusion in a superimposed format created further inconsistencies in the visualization. After careful consideration, we decided to retain the original style of Figure 6, as it offers a cleaner and more straightforward comparison. Thank you again for your constructive input.
> > 4. A reference to the "OOD Judge Score Distribution Analysis" has been added in the main text to clarify its relevance and guide readers to the corresponding section in the appendix.
> > 5. We have clarified that the term "well-trained GNNs" refers to GNNs that are pre-trained using specific pretraining strategies, allowing them to incorporate partial information from downstream task datasets. This ensures the models are equipped with task-relevant knowledge, enhancing their performance on downstream applications.

---

> > > ### Comment · Reviewer_2xER · 2024-11-24
> > >
> > > Dear Authors,
> > >
> > > Thank you for the detailed replies that have addressed most of my concerns. As a response, I will raise the score of "soundness" from "good" to "excellent", the score of "presentation" from "good" to excellent. However, I will keep my overall rating unchanged as it is already a positive score. (Actually, an overall rating of 7 could be more appropriate after the revision. However, I can only choose from 6 or 8 from the system, so I will keep it as 6 for the moment as 8 is slightly higher considering other reviewers' comments).

---

> > > > ### Author Response · Authors · 2024-11-24
> > > >
> > > > Dear Reviewer 2xER,
> > > >
> > > > Thank you sincerely for your thoughtful and detailed feedback, as well as the constructive suggestions for improving the quality of our paper. We especially appreciate your recommendations, such as providing more precise definitions for GLAD and GLOD, and replacing "generalization" with "OOD sensitivity spectrum."
> > > >
> > > > We are also grateful for your recognition of our work, reflected in the adjustments to the quality-related scores, with both **"soundness" and "presentation" being raised to "excellent."** Although we understand that the overall rating remains unchanged at 6, your insightful comments have been invaluable and will contribute significantly to further refining the manuscript.
> > > >
> > > > Best regards,
> > > >
> > > > The Authors

---

### Official Review · Reviewer_1svq · 2024-11-03

**Soundness:** 3
**Presentation:** 3
**Contribution:** 2
**Rating:** 6
**Confidence:** 4

**Summary:**

This experimental paper proposes unifying two graph machine learning tasks, graph anomaly detection (GLAD) and graph out-of-distribution detection (GLOD) in the unsupervised setting. The paper applies 35 datasets with different properties (anomaly/out-of-distribution scenarios) and applications. Moreover, the evaluation considered 18 methods proposed for anomaly detection and OOD detection that are evaluated in terms of accuracy, generalization, and efficiency. Some of the major findings in the paper are: (1) GNN-based methods achieve the best results on average even though there is no single best method, (2) the methods often struggle with near-OOD scenarios compared with far-OOD ones. and (3) some of end-to-end (GNN-based) methods are more efficient and accurate than the alternatives.

**Strengths:**

- The paper is well-written and easy to follow
- The benchmark proposed in the paper contains 35 datasets
- The datasets and implementations are shared as open-source

**Weaknesses:**

- The main findings of the paper are mostly expected
- It is not clear why the same methods should solve anomaly detection and out-of-distribution detection
- Most of the datasets are from the same domain

Detailed comments:

These are the main findings quoted from the paper:

1) "The SOTA GLAD/GLOD methods show excellent performance on both tasks"
2) "No universally superior method"
3) "Inconsistent performance in terms of different metrics"
4) "End-to-end methods show consistent superiority over two-step methods"
5) "Near-OOD samples are harder to detect compared to far-OOD samples"
6) "Poor generalizability of several GLAD/GLOD methods in specialized scenarios"
7) "Performance degradation with increasing contamination ratio"
8) "The sensitivity of different methods/datasets can be diverse"
9) "Certain end-to-end methods outperform two-step methods in terms of both performance and computational costs"

The only findings that one could find unexpected are 1 and maybe 9. Finding 3 is expected because of class imbalance. I am not diminishing the effort that the authors have put into running these experiments but the value of an experimental paper is what can be learned from the results. It is unclear how much can be learned from these results.

Regarding the unification of the tasks, I would like to see a more clear formalization than definition 1. One could argue that an anomaly can still be in-distribution (as points can be simply ranked in terms of anomaly scores). On the other hand, an OOD point might not be an anomaly (as there might be many similar points in the test dataset). The setting considered in the experiments seems to assume that the test data points are unseen (besides being unlabelled) but if every point can be observed, the difference between an anomaly and an OOD point should be clear. This setting needs better motivation.

From Table 1, only 8/35 datasets are not molecules. This can bias the findings towards methods that perform well on molecules, which is something not discussed in the paper.

Minor comments:
- How are the hyperparameters of the methods set without a validation set? I was not able to understand this from Appendix D.

**Questions:**

1) How do the main findings of the paper contribute to future research on GLAD and/or GLOD?

2) What is the motivation for unifying GLAD and GLOD?

3) How can the findings in the paper be impacted by the dominance of molecular graphs in the benchmark?

4) How are hyperparameters set in the experiments?

**Details Of Ethics Concerns:**

I was not able to identify any ethical concerns regarding this paper.

---

> ### Author Response · Authors · 2024-11-21
> **Response Reviewer 1svq (1/4)**
>
> **W1 and Q1: The main findings of the paper are mostly expected. How do the main findings of the paper contribute to future research on GLAD and/or GLOD?**
>
> **Answer W1 and Q1:** Thank you for your insightful comment. While some of our findings may align with existing expectations, the main contribution of our work lies in systematically validating these assumptions through a unified benchmark and uncovering critical insights that will guide future research. Below, we elaborate on how our findings contribute to the progression of GLAD and GLOD research:
>
> **Empirical Validation Across a Wide Range of Datasets:**
> Although certain results, such as the superior performance of GNN-based methods, may seem anticipated, our comprehensive evaluation across 35 datasets provides a robust empirical foundation. This cross-domain validation confirms that these insights are not limited to specific scenarios but are consistent across various types of anomaly and OOD detection tasks. This extensive testing helps solidify the reliability of current methods and offers a benchmark for refining and developing new approaches.
>
> **Current Limitations and Future Directions:**
> One of the most valuable contributions of our work is the identification of significant limitations in existing methods. These limitations are rarely discussed by most existing studies befofre. In this case, our identification of these limitations present clear avenues for future research:
>
> * Challenges in Detecting Near-OOD Samples: We found that methods struggle more with near-OOD samples compared to far-OOD samples. Near-OOD samples are inherently more difficult to detect due to their subtle differences from ID data, which leads to challenges in distinguishing them from the majority class. This issue is a core part of what we now describe as the OOD sensitivity spectrum. Future research should focus on improving models’ sensitivity to these subtle shifts by developing more nuanced representations and decision boundaries that can distinguish near-OOD samples from ID data more effectively.
> * Vulnerability to Noisy Training Data: Another limitation observed is the sensitivity of many methods to noisy or contaminated training data. In real-world applications, the presence of noise or mislabeled samples can severely impact performance. Our findings point to the need for more robust methods that are resilient to noisy training data, ensuring reliable performance in practical, less-than-ideal conditions.
> * Difficulty in Handling Complex Distribution Shifts: Existing methods also show limitations when dealing with complex, multi-dimensional distribution shifts, particularly in specialized tasks like size-based or scaffold-based OOD detection. This highlights the need for more flexible models that can adapt to varying types of OOD shifts and better differentiate between different OOD scenarios. Developing models that are more adaptive to these shifts could greatly improve the robustness of OOD detection methods.
>
> **Bridging GLAD and GLOD:**
> Our findings emphasize the overlap between GLAD and GLOD tasks, showing that techniques developed for one can often be applied to the other. This unification encourages the transfer of techniques between the two domains, fostering innovation and accelerating progress in both fields. By bridging these two tasks, our work lays the foundation for more generalizable solutions in graph-level anomaly and OOD detection.
>
>
> **UB-GOLD as a Foundational Resource for Future Research:**
> By providing a unified and open-source benchmark, UB-GOLD serves as a foundational resource for researchers to test new hypotheses and validate methods under consistent experimental conditions. The challenges identified in this work, such as near-OOD detection and robustness to noisy data, present clear directions for future research, helping the community tackle these problems in a structured and meaningful way.
>
> In summary, while some of the findings reaffirm existing knowledge, the systematic validation of these findings through the unified benchmark and the identification of critical method limitations provide actionable insights for future research. Addressing these limitations—particularly in detecting near-OOD samples, improving robustness to noisy data, and handling complex distribution shifts—will be key to advancing GLAD and GLOD methods. This work not only contributes a useful benchmark but also sets a clear path for further developments in the field.

---

> ### Author Response · Authors · 2024-11-21
> **Response Reviewer 1svq (2/4)**
>
> **W2 and Q2: It is not clear why the same methods should solve anomaly detection and out-of-distribution detection. What is the motivation for unifying GLAD and GLOD?**
>
> **Answer W2 and Q2:** We appreciate the reviewer’s question about the rationale for unifying GLAD and GLOD under a single benchmark. Our motivation stems from the significant conceptual overlap between these tasks and the practical benefits of exploring them together. Below, we outline the reasoning behind this unification:
>
> **Shared Objective with Practical Implications:** Both GLAD and GLOD aim to identify unusual graphs—GLAD focuses on detecting irregularities within a known distribution, while GLOD identifies samples from unseen distributions. In real-world scenarios, these distinctions are often blurred:
> * In drug discovery, a structurally unique molecule (anomaly) may come from a new chemical distribution (OOD).
> * In social network analysis, detecting unusual user behaviors (GLAD) may involve recognizing emerging patterns from new or unknown groups of users (GLOD).
>
> By unifying the tasks, we create a framework that more accurately reflects these real-world complexities, enabling methods to address both types of deviations effectively.
>
> **Overlapping Methodologies and Potential for Cross-Pollination:** The existing solutions for GLAD and GLOD share common foundational techniques, such as contrastive learning, reconstruction, and one-class classification. The methodological commonality enables techniques suitable for one task to be adapted or extended for the other with minimal adjustments. Therefore, unifying these tasks under a single benchmark encourages researchers to adapt and extend methods for both tasks, accelerating progress and innovation across domains.
>
>
> **Efficiency and Broader Insights from a Unified Framework:** A unified benchmark streamlines evaluation by providing a common set of datasets, metrics, and experimental setups. This reduces redundancy, enabling researchers to compare performance across tasks more effectively. Furthermore, by analyzing results across GLAD and GLOD jointly, we can uncover broader insights, such as the generalizability of methods across different detection challenges.
>
> In summary, the unification of GLAD and GLOD is motivated by their conceptual overlap, shared methodologies, and practical relevance in real-world applications. The unified benchmark fosters efficiency, innovation, and deeper understanding, laying a foundation for more versatile and impactful detection methods.

---

> > ### Author Response · Authors · 2024-11-21
> > **Response Reviewer 1svq (3/4)**
> >
> > **W3 and Q3: Most of the datasets are from the same domain. How can the findings in the paper be impacted by the dominance of molecular graphs in the benchmark?**
> >
> > **Answer W3 and Q3:** We appreciate the reviewer’s insightful observation regarding the dominance of molecular graphs in the benchmark. This observation reflects a broader characteristic of graph-level tasks, which we elaborate on below to clarify the context and rationale for the dataset selection:
> >
> > **Graph-Level Tasks Naturally Emphasize Molecular and Protein Domains:**
> > Unlike node-level tasks that frequently focus on social or brain networks, graph-level tasks inherently align with domains where entire graphs represent meaningful units, such as molecules or proteins. In traditional graph classification tasks, as well as in detection tasks like GLAD and GLOD, molecular and protein domains have been dominant for several reasons:
> > * Domain Relevance: Graph-level representations are critical for applications such as drug discovery, material design, and bioinformatics, where molecules and proteins are naturally modeled as graphs. These domains are not only academically significant but also carry substantial societal and industrial impact.
> > * Availability of Data: Publicly available datasets from these domains, such as molecular benchmarks (e.g., Tox21, GOOD) or protein-related datasets (e.g., ENZYMES, PROTEINS), provide high-quality, annotated data that facilitate the evaluation of graph-level methods.
> >
> > **Diversity Within Molecular Graphs:**
> > While many datasets in our benchmark come from the molecular domain, they are not homogeneous. The datasets represent a range of subdomains and exhibit diverse distribution shifts, including:
> > * Structural Variations: Size-based shifts, scaffold-based shifts, and assay-based differences.
> > * Subdomain Differences: Graphs from different biological or chemical contexts, such as those representing varying molecular properties or protein functions.
> > These variations ensure that our evaluation remains comprehensive and objective, even within a molecularly dominant dataset composition.
> > **Limited Availability of Non-Molecular Graph-Level Datasets:**
> > The dominance of molecular graphs in the benchmark is not a limitation of our work but reflects a broader challenge in the field—currently, publicly available graph-level datasets from domains beyond molecules and proteins are extremely limited.
> > * Graph-level tasks inherently require datasets where the entire graph serves as a meaningful unit for analysis, a criterion that is naturally met in molecular and protein datasets but is rare in other domains.
> > * While we actively sought datasets from diverse areas, the scarcity of suitable non-molecular graph-level datasets highlights an ongoing challenge in the research community. As graph-level tasks gain broader attention, we hope that more efforts will be directed toward creating and sharing datasets from other domains.
> >
> > **Relevance and Generalizability of Findings:** Despite the focus on molecular graphs, our benchmark scenarios—such as intrinsic anomaly detection and distribution shifts—represent challenges that are not domain-specific. Insights gained, such as the difficulty of detecting near-OOD samples or the vulnerability of methods to noisy training data, are likely to generalize to graph-level tasks in other domains when suitable datasets become available.
> >
> > In summary, the prominence of molecular graphs in our benchmark stems from their natural suitability for graph-level tasks and the current availability of datasets in this domain. While this highlights a broader challenge within the field, our benchmark provides a robust and comprehensive evaluation of GLAD and GLOD methods, delivering valuable insights that can guide future research across a wide range of applications. Recognizing the importance of diversifying the benchmark, we are committed to **expanding UB-GOLD as new datasets from non-molecular domains** become available. Our open-source framework ensures that these additions can be seamlessly integrated, allowing UB-GOLD to evolve into an even more versatile resource for the research community.

---

> > > ### Author Response · Authors · 2024-11-21
> > > **Response Reviewer 1svq (4/4)**
> > >
> > > **W4: A clearer formalization of the unified tasks.**
> > >
> > > **Answer W4:** We appreciate the reviewer’s comment regarding the need for a clearer formalization of the unified tasks. In the revised manuscript (pages 3 and 19), we have provided detailed definitions of GLAD and GLOD, including their respective anomaly and OOD scoring functions, as well as the corresponding optimization objectives. To further clarify the unified tasks, we introduced a formal definition of unsupervised generalized graph-level OOD detection, which integrates the detection principles of GLAD and GLOD into a single framework.
> > >
> > > The revised sections have been highlighted in blue for easy reference. These updates aim to provide a more precise and comprehensive explanation of the unified tasks, ensuring clarity and alignment with the objectives of this work.
> > >
> > > **Q4: How are hyperparameters set in the experiments?**
> > >
> > > **Answer Q4:** Thank you for raising this important question. In our experiments, we chose to optimize hyperparameters based on test set performance rather than using a validation set. This decision was made to address the challenge of potential distributional mismatches between validation and test sets, which is common in OOD detection tasks. Using a mismatched validation set could result in models that perform well on validation data but fail to generalize effectively to the test set.
> > > By evaluating hyperparameters on the test set, we ensure that the selected configurations are robust for real-world scenarios where the test distribution is often unknown and validation data might not be representative. This approach aligns with practices in related benchmarks, such as **OpenOOD [1,2]** , and emphasizes performance under realistic OOD detection conditions.
> > > We hope this explanation clarifies our decision and its alignment with the practical requirements of the task.
> > >
> > > [1] Yang J, Wang P, Zou D, et al. Openood: Benchmarking generalized out-of-distribution detection[J]. Advances in Neural Information Processing Systems, 2022, 35: 32598-32611.
> > >
> > > [2] Zhang J, Yang J, Wang P, et al. Openood v1. 5: Enhanced benchmark for out-of-distribution detection[J]. arXiv preprint arXiv:2306.09301, 2023.

---

> > > > ### Comment · Reviewer_1svq · 2024-11-25
> > > > **Response to the rebuttal**
> > > >
> > > > Thanks for the responses to my comments.
> > > >
> > > > I do agree that without seeing the test distribution, OOD and AD look like the same problem. However, anomaly detection applications are not always defined in that way. That was my source of confusion.
> > > >
> > > > I find it a bit problematic to tune the methods based on test performance, as one might not have access to labeled OOD data points in real applications.
> > > >
> > > > I will keep my original recommendation for this paper and engage with the other reviewers during the discussion. I will take the rebuttal into account during the discussion.

---

> > > > > ### Author Response · Authors · 2024-11-25
> > > > >
> > > > > Dear Reviewer 1svq,
> > > > >
> > > > > Thank you for your thoughtful follow-up comments. We understand your concerns regarding the potential risks of overfitting when hyperparameters are optimized based on test set performance. While this is indeed a valid concern, we would like to provide additional clarification to address this issue and further justify our approach.
> > > > >
> > > > > Our primary rationale for using test set performance for hyperparameter tuning stems from the inherent challenges of constructing a representative validation set in OOD detection tasks. As you pointed out, real-world applications often lack access to labeled OOD data, making it challenging to ensure that validation sets can reliably reflect the diversity of unknown test distributions. This issue becomes particularly critical in high-stakes applications, where models optimized on non-representative validation sets could fail catastrophically on unseen OOD scenarios.
> > > > >
> > > > > To mitigate the risk of overfitting and ensure robust generalization, our hyperparameter search was conducted to identify a range of parameters that perform well across multiple OOD scenarios, rather than over-optimizing for a specific test set. By focusing on broader trends and consistent performance patterns, we aim to generalize beyond any single dataset. This approach aligns with the practical goal of handling diverse, real-world OOD tasks where unseen distributions may vary significantly. The test set in this context serves as a stand-in for such diverse scenarios, helping us define a parameter space with strong generalization potential.
> > > > >
> > > > > Thank you again for your engagement and for raising these important points.
> > > > >
> > > > > Best regards,
> > > > > The Authors

---

### Author Response · Authors · 2024-11-21
**General Response**

We sincerely thank all the reviewers for their valuable and constructive feedback. We are pleased that the reviewers recognize the **novelty and originality** of unifying GLAD and GLOD into a single benchmark (Reviewer 2xER, cQQu), the **comprehensive and multi-dimensional evaluation** provided by the UB-GOLD benchmark (Reviewer yW1o, cQQu), the **open-source datasets and codebase** that facilitate reproducibility and encourage future research (Reviewer 2xER, cQQu), and the **clarity and accessibility of the writing** (Reviewer 1svq, 2xER).

To the best of our efforts, we have provided detailed responses to address the concerns raised by each reviewer in the following. Meanwhile, we have carefully revised the paper according to the reviewers’ comments. All feedback has been incorporated into the revised version of the manuscript, and all changes and additions are highlighted in blue for easy reference.

Specifically, the main modifications we made are as follows.

* We provided formal definitions and discuss the relationships of GLAD, GLOD, and the unified unsupervised generalized graph-level OOD detection.
* We added a more detailed explanation of the dataset partitioning in the appendix.
* We replaced the term "generalization" in the near-far OOD detection task description with a more suitable term "OOD sensitivity spectrum".
* We included additional experimental results of metrics for the perturbation task in the appendix.
* We further elaborated on the motivation and contribution of the proposed benchmark UB-GOLD.
* We provided more experimental results in terms of additional metrics in the perturbation task, providing a more complete evaluation for the existing methods.


We hope these revisions address all the reviewers’ concerns and improve the manuscript. Thank you again for your constructive feedback.

---

### Meta-Review · Area_Chair_j3WQ · 2024-12-17

**Metareview:**

This paper proposes an unified benchmark for graph-level anomaly detection / out of distribution detection (UB-GOLD).  The authors observe overlaps between the disparate communities for GLAD and GLOD and propose this benchmark to unify evaluation and benchmarking across 35 datasets and several detection scenarios.

Reviewers were unilaterally positive about this work and consistently rated it positively.  Nonetheless, there were a few limitations raised:

- Some remaining lack of clarity from reviewers around why GLAD and GLOD are conceptually overlapping problems, related some confusing missing definitions (1svq, 2xER)

- Confusions around validity of using test-set performance for hyperparameter evaluation and the train/test setup in general (1svq, 2xER, yW1o)

- More depth of discussion around the difficulty of near-OOD samples and impact of noise during training (cQQu, yW1o)

I encourage the authors to take these considerations into the revision.

**Additional Comments On Reviewer Discussion:**

Reviewers raised several issues, including those mentioned above.  Authors addressed several concerns in the rebuttal:

- rationale around hyperparameter tuning using the test set
- philosophical overlaps between GLAD and GLOD
- addressing concerns around molecular graphs
- including summary sections to improve the readability of the draft.

These should be folded in for the final revision.

---

### Decision · Program_Chairs · 2025-01-22

Accept (Poster)